# TwinVLA: Data-Efficient Bimanual Manipulation with Twin Single-Arm Vision-Language-Action Models

**Hokyun Im**[1,2]   **Euijin Jeong**[1]   **Andrey Kolobov**[2]   **Jianlong Fu**[2]   **Youngwoon Lee**[1]
[1]Department of Artificial Intelligence, Yonsei University   [2]Microsoft Research
https://jellyho.github.io/TwinVLA/

## Abstract

Vision-language-action models (VLAs) trained on large-scale robotic datasets have demonstrated strong performance on manipulation tasks, including bimanual tasks. However, because most public datasets focus on single-arm demonstrations, adapting VLAs for bimanual tasks typically requires substantial additional bimanual data and fine-tuning. To address this challenge, we introduce TwinVLA, a modular framework that composes two copies of a pretrained single-arm VLA into a coordinated bimanual VLA. Unlike monolithic cross-embodiment models trained on mixtures of single-arm and bimanual data, TwinVLA improves both data efficiency and performance by composing pretrained single-arm policies. Across diverse bimanual tasks in real-world and simulation settings, TwinVLA outperforms a comparably-sized monolithic RDT-1B model without requiring *any* bimanual pretraining. Furthermore, it narrows the gap to state-of-the-art model $\pi_0$, which relies on extensive proprietary bimanual data and compute cost. These results establish our modular composition approach as a data-efficient and scalable path toward high-performance bimanual manipulation, leveraging public single-arm data.

## 1 Introduction

Thanks to publicly available large-scale robotic datasets, vision-language-action models (VLAs) have shown impressive performance in single-arm robotic manipulation, effectively adapting to downstream tasks and generalizing across diverse tasks, objects, and environments (Zitkovich et al., 2023; Open X-Embodiment Collaboration et al., 2024; Kim et al., 2024; Black et al., 2024). However, extending these successes to *bimanual* manipulation remains challenging, as public bimanual datasets are scarce, and existing approaches often rely on large, proprietary datasets that require thousands of hours of data collection and curation (Black et al., 2024), limiting reproducibility and progress.

Can we build strong bimanual VLAs without collecting or fine-tuning on large bimanual datasets by leveraging existing single-arm data? In this work, we propose a highly data-efficient adaptation paradigm for bimanual control that eliminates the need for prohibitive bimanual pretraining. By effectively repurposing a single-arm VLA, we demonstrate that complex bimanual skills can be mastered using only minimal target-domain demonstrations, establishing a practical and reproducible pathway to bimanual manipulation.

To effectively realize this transfer of single-arm priors to bimanual control, the choice of underlying architecture is critical. Recent cross-embodiment learning work typically trains monolithic models on multi-robot datasets (Open X-Embodiment Collaboration et al., 2024) employing embodiment-specific action decoders (Octo Model Team et al., 2024; NVIDIA et al., 2025) or shared, zero-padded action spaces (Liu et al., 2024; Black et al., 2024). Although promising, differences in observation and action spaces introduce heterogeneity, forcing a single model to handle disparate action spaces, and monolithic training underutilizes the *modular* structure inherent to bimanual tasks.

A *modular* perspective on bimanual manipulation is supported by neuroscience: human bimanual manipulation is the coordination of arm-specific motor primitives rather than a single monolithic controller. Dedicated neural circuits, such as the Supplementary Motor Area (SMA) and the corpus callosum, orchestrate and synchronize the two arms (Sadato et al., 1997; Swinnen, 2002). Similar

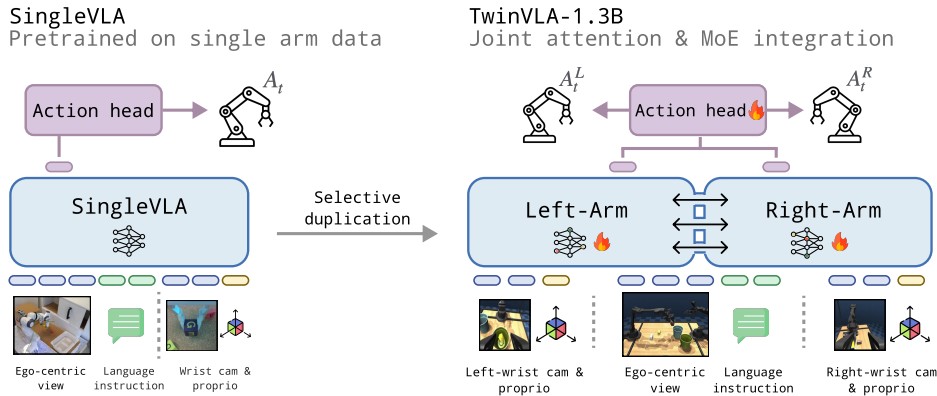

Figure 1: **Overview of TwinVLA.** Inspired by humans' two-arm coordination for bimanual manipulation, TwinVLA duplicates a VLM backbone pretrained on cross-embodiment single-arm data (*Left*) to form two arm-specific branches linked via **Joint Attention** (*Right*). Shared inputs (ego-centric views, language instructions) are routed via a mixture-of-experts (MoE) to improve computational efficiency. Only the VLM backbone is duplicated, keeping the increase in model size minimal.

principles have proven effective in vision-language modeling, where interaction between modality-specific backbones improves its efficiency and effectiveness (Liang et al., 2024).

Inspired by these insights, we propose TwinVLA, a modular architecture that operationalizes this coordination-centric view. Instead of training from scratch, TwinVLA leverages a pretrained single-arm VLA. Specifically, we first design a lightweight, compact single-arm VLA, which we call SingleVLA (Appendix A). We pre-train a 0.8B-size SingleVLA for single-arm manipulation on the OXE dataset (Open X-Embodiment Collaboration et al., 2024). We then duplicate this SingleVLA and integrate the two "twin" instances through a lightweight coordination method. This design is highly data-efficient: it eliminates the need for a bimanual pretraining dataset and achieves strong performance with only a small amount of bimanual demonstrations for fine-tuning.

To integrate two SingleVLAs into a bimanual policy, TwinVLA utilizes a joint attention (Liang et al., 2024) across the twin models, as illustrated in Figure 1. This allows the twin SingleVLAs to exchange information and coordinate their actions, while preserving their pretrained capabilities. This approach is made feasible without significant overhead, as we duplicate only the VLM backbone and utilize a Mixture-of-Experts (MoE) to efficiently manage shared inputs. In contrast to monolithic cross-embodiment models (Liu et al., 2024; Octo Model Team et al., 2024; Doshi et al., 2024), our approach yields better performance and data efficiency, significantly reducing the need for large-scale bimanual data collection and compute.

We evaluate TwinVLA across a broad range of environments, including a complex, long-horizon real-world task and a diverse suite of bimanual manipulation tasks in simulations. Despite leveraging only public single-arm data and limited bimanual fine-tuning data, TwinVLA achieves performance comparable to state-of-the-art bimanual policies.

In summary, our main contributions are threefold:

- We propose a novel modular architecture for bimanual manipulation that integrates two copies of a pretrained SingleVLA with a lightweight coordination method based on joint attention with MoE, enabling synchronized two-arm control.
- We present a data-efficient paradigm that adapts our twin architecture into a capable bimanual policy for a target task by fine-tuning on only a small bimanual dataset, crucially without requiring additional pretraining, thereby eliminating the need for large-scale bimanual data.
- Through extensive experiments across real and simulated bimanual tasks, TwinVLA matches or surpasses state-of-the-art models trained on far larger bimanual data and compute.

Together, these findings identify our modular SingleVLA composition approach as a scalable, efficient path to high-performance bimanual manipulation.

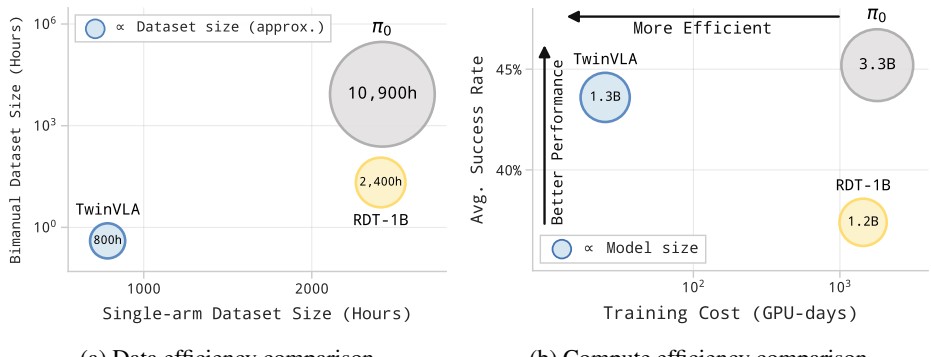

(a) Data efficiency comparison       (b) Compute efficiency comparison

Figure 2: **(a) Data efficiency.** TwinVLA requires only $\sim 800h$ of single-arm and 50 episodes of target bimanual data, significantly less than RDT-1B ($\sim 2,400h$) and $\pi_0$ ($\sim 10,900h$) in total. **(b) Compute efficiency.** RDT-1B and $\pi_0$ require high compute (exceeding $1,000$ H100 GPU-days), whereas TwinVLA achieves higher or comparable performance with only 25 H100 GPU-days.

## 2 RELATED WORK

Bimanual manipulation policies are essential to enable robots to perform complex tasks that require coordinated two-handed control, such as folding laundry (Bersch et al., 2011; Avigal et al., 2022), assembling parts (Stavridis & Doulgeri, 2018), or wiping the plate (Black et al., 2025; Chi et al., 2024b). Learning effective bimanual policies is challenging due to high-dimensional, tightly coupled action spaces and the scarcity of high-quality bimanual demonstrations (Lee et al., 2020; Xie et al., 2020). Consequently, specialist methods, such as Diffusion policy (Chi et al., 2024a) and ACT (Zhao et al., 2023), trained only on target-task demonstrations, struggle on precise, long-horizon tasks.

Recent works have explored various architectures for bimanual control to explicitly model the inter-dependencies between arms (Lee et al., 2024; Kobayashi & Buamanee, 2025), or focus on high-level language planning via VLM (Gbagbe et al., 2024). Anybimanual (Lu et al., 2025) introduces a high-level skill manager to coordinate primitives and visual aligner to mask 3D voxels for decoupled policies, benefiting from both high-level managing and architectural inductive bias. While promising, it is difficult to generalize these methods, as they are often limited to small-scale scenarios, handle only low-dexterity tasks, or backbone constraints (Grotz et al., 2024; Shridhar et al., 2022).

Alternatively, another line of research extends successful unimanual Vision-Language-Action (VLA) models (Liu et al., 2023b; Zitkovich et al., 2023; Li et al., 2024b) to bimanual tasks. This transition is challenging due to the scarcity of bimanual data, as public datasets are predominantly unimanual. To overcome this, prior work trains 'monolithic' models, requiring large-scale bimanual data collection and intensive pretraining. For example, RDT-1B (Liu et al., 2024) required massive pretraining and fine-tuning (reportedly a month on 48 H100 GPUs), and $\pi_0$ (Black et al., 2024) relies on a $10,000$-hour proprietary dataset, both incurring high computational costs. Furthermore, the proprietary nature of these datasets limits reproducibility and broader adoption.

In contrast to both monolithic, compute-heavy pretraining and specialized architectural designs, our approach adopts a modular, coordination-centric design. While Anybimanual (Lu et al., 2025) introduces novel inductive biases for coordination, these are often difficult to integrate into general-purpose VLA frameworks due to specific backbone constraints. Our method, however, is designed to leverage and scale the existing generalist VLAs. We first train a SingleVLA on large-scale public single-arm data, duplicate to couple them, and then fine-tune it on bimanual tasks—allowing each stage to benefit from the most suitable data (see Figure 2). This composition-based approach avoids bimanual pretraining, requires only a small amount of bimanual fine-tuning, better preserves the strong capabilities of single-arm policies, and significantly improves data and compute efficiency.

## 3 PRELIMINARIES

This paper aims to develop a data-efficient framework for learning bimanual manipulation policies by building upon pretrained single-arm Vision-Language-Action (SingleVLA) models. This section

formalizes the single-arm and bimanual settings, briefly describes the VLA training objective, and introduces the core architectural concepts we leverage.

## 3.1 FORMULATING THE BIMANUAL VLA POLICY

Our goal is to extend a pretrained SingleVLA $\pi_{\text{single}}$ into a bimanual policy $\pi_{\text{twin}}$ applicable to target bimanual tasks. A VLA $\pi(A_t \mid o_t)$ predicts an *action chunk* $A_t = (a_t, a_{t+1}, \ldots, a_{t+T-1})$ of length $T$ from an observation $o_t$. For single-arm manipulation, the observation $o_t^{\text{single}} = ((l, I_{\text{ego}})_t, (I_{\text{wrist}}, d)_t)$ includes a language prompt $l$, an ego-centric image $I_{\text{ego}}$ (shared input), and an arm-specific wrist image $I_{\text{wrist}}$ with proprioception $d$ (arm-specific input). We train $\pi_{\text{single}}(A_t \mid o_t^{\text{single}})$ to predict the action chunk for one arm. For bimanual manipulation, the observation aggregates both right $(R)$ and left $(L)$ arm-specific input, $o_t^{\text{twin}} = ((l, I_{\text{ego}})_t, (I_{\text{wrist}}^R, d^R)_t, (I_{\text{wrist}}^L, d^L)_t)$, and the policy $\pi_{\text{twin}}(A_t^R, A_t^L \mid o_t^{\text{twin}})$ outputs a joint action chunk for right and left arms.

## 3.2 TRAINING VLAS WITH CONDITIONAL FLOW MATCHING

We train our VLA models to predict continuous robot actions from observations. Each observation $o_t$ is tokenized and fed into the VLM backbone to produce an output embedding $h_t$ (from a learnable readout token $r_t$). To enable continuous action prediction from $h_t$, we attach an action head $v_\theta(A_t^\tau, h_t, d_t)$ and train it using a conditional flow matching objective. The action head is trained with the following loss function:

$$\mathcal{L}^T(\theta) = \mathbb{E}_{p(A_t|o_t),q(A_t^\tau|A_t)} \|v_\theta(A_t^\tau, h_t, d_t) - \mathbf{u}(A_t^\tau \mid A_t)\|^2, \tag{1}$$

where $h_t$ is the VLM output embedding and $d_t$ is proprioception. This objective trains the action head $v_\theta$ to predict the reference flow $\mathbf{u}$ from a noised action chunk $A_t^\tau$ to the target action chunk $A_t$, conditioned on the VLM output and proprioception.

During inference, we sample actions using the forward Euler integration method. Starting from $A_0 \sim N(0, I)$, we iteratively update the action using the learned flow $v_\theta$:

$$A_t^{\tau+\delta} = A_t^\tau + \delta v_\theta(A_t^\tau, h_t, d_t), \tag{2}$$

where we set the sampling step $n = 10$ and use $\delta = \frac{1}{n}$.

## 3.3 MIXTURE-BASED ARCHITECTURES

To adapt Transformers for multi-modal inputs, various mixture-based architectures have been explored and shown to be effective. These approaches range from combining entire, modality-specific backbones to ensembling or mixing individual layers within a single backbone. We briefly introduce two such paradigms that inform our design: a *model-level* Mixture-of-Transformers (MoT), which coordinates separate backbones, and a *layer-level* Mixture-of-Experts (MoE), which enables efficient, sparse computation.

The MoT architecture (Liang et al., 2024) enables efficient information sharing between separate, modality-specific backbones (e.g., text and image). It introduces *joint attention*, a shared self-attention layer performed over the union of multimodal inputs, allowing each modality to directly attend to the others. Meanwhile, modality-specific components such as feed-forward networks remain separate, making fusion lightweight yet effective.

MoE (Shazeer et al., 2017) scales model capacity efficiently by routing each input $x$ through a weighted combination of expert feed-forward networks using a gating function, yielding $\text{MoE}(x) = \sum_i w_i E_i(x)$, where $w_i$ denotes the routing weight.

## 4 TWINVLA

TwinVLA is a modular architecture that transforms a pretrained single-arm VLA into a coordinated bimanual policy. The overall computation flow of our architecture is described in Algorithm 1 and Figure 3. TwinVLA integrates single-arm policies through three core principles: (1) selective module duplication (Section 4.1), (2) cross-arm fusion via joint attention (Section 4.2), and (3) efficient shared representation via Mixture-of-Experts (Section 4.3).

---

**Algorithm 1** TwinVLA

---

$X_0^m$: encoded inputs from $o_t^{\text{twin}}$ (Section 3.1) for each input $m \in \{\text{shared}, \text{left}, \text{right}\}$.
$\text{FFN}_b$: feed-forward network layer from each backbone $b \in \{\text{left}, \text{right}\}$.
$N$: Number of transformer layers
**for** $n = 0$ to $N - 1$ **do**                             ▷ Iterate every transformer layer
    // Prepare Q, K, V for each input $m$
    **for** each input $m \in \{\text{shared}, \text{left}, \text{right}\}$ **do**
        $Q_n^m, K_n^m, V_n^m \leftarrow \text{Norm}(\text{Proj}(X_n^m))$         ▷ Input-specific projections, Algorithm 3
    **end for**
    // Joint attention across inputs with attention re-weighting
    $\{A_n^m\} \leftarrow \text{JointAttention}(\{Q_n^m\}, \{K_n^m\}, \{V_n^m\}, M)$     ▷ Algorithm 2, with mask $M$ Figure 3a
    // Residual & FFN / MoE
    **for** each input $m \in \{\text{shared}, \text{left}, \text{right}\}$ **do**
        $H_n^m \leftarrow X_n^m + \text{Norm}(\text{Proj}(A_n^m))$         ▷ Input-specific output projection, Algorithm 3
        $F_n^m \leftarrow \text{MoE}(H_n^m)$ **if** $m = \text{shared}$ **else** $\text{FFN}_m(H_n^m)$     ▷ MoE for shared input, Equation (3)
        $X_{n+1}^m \leftarrow H_n^m + \text{Norm}(F_n^m)$         ▷ Residual connection with norm, Algorithm 3
    **end for**
**end for**
**return** $\{X_N^m\}$              ▷ Return outputs, this will be used for action decoding

---

## 4.1 SINGLE-ARM POLICY DUPLICATION

We first pre-train a VLA on a single-arm dataset, which we refer to as SingleVLA. Note that existing pre-trained models can also be used for this purpose. To construct TwinVLA from SingleVLA, we initialize the twin policies for the left and right arms by copying the pretrained SingleVLA. However, instead of duplicating the full model, we share the vision encoder and DiT (Peebles & Xie, 2023) action head while fully replicating the VLM. Each arm has its own lightweight proprioception encoder. This design yields a compact $1.3$B-parameter model, comparable to the $1.2$B-parameter RDT-1B, without significantly increasing computational cost.

Visual inputs are processed by the shared encoder, and each VLM produces readout tokens that are jointly decoded by the shared DiT. This design is motivated by the principle that general visual understanding (image encoding) and low-level motor control (action decoding) are largely embodiment-agnostic skills that can be effectively shared for both arms. In contrast, the VLM, which decides output action given encoded observation, is fully replicated to allow for specialized control.

## 4.2 JOINT ATTENTION FOR CROSS-ARM FUSION

We integrate arm-specific inputs using a Joint Attention mechanism inspired by MoT (Liang et al., 2024). As illustrated in Figure 3b and Algorithm 1, this is achieved by sharing only the self-attention layers across the VLM backbones. Specifically, we concatenate the $Q, K, V$ from both backbones, perform self-attention, and subsequently split the outputs back to their respective streams, while other components such as projections use arm-specific networks from each arm's VLM backbone. Unlike $\pi_0$ (Black et al., 2024), which links a VLM with an action head, we connect two VLMs directly. We elaborate joint attention mechanism in detail on Algorithm 2.

**Causal joint attention mask.** Effective joint attention requires appropriate attention masking. Standard LLMs use a lower-triangular attention mask for causal prediction. To support joint attention among the shared and arm-specific inputs, we designed the attention mask for TwinVLA as shown in Figure 3a. Specifically, we embed lower-triangular masks within each arm's region while treating the shared modality as fully accessible. Each arm also attends to half of the other's tokens, enabling symmetric cross-arm interaction without violating autoregressive constraints.

## 4.3 MIXTURE-OF-EXPERTS INTEGRATION

In TwinVLA, feeding shared inputs $(l, I_{\text{ego}})_t$ redundantly to both VLMs significantly increases VRAM usage. To address this, we process shared tokens as a single sequence by employing a MoE

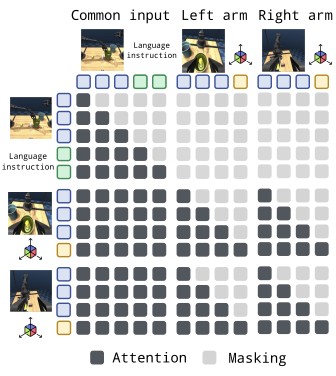
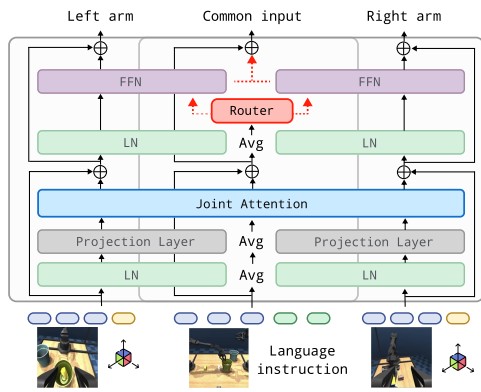

(a) Causal joint attention mask · (b) Transformer block of TwinVLA

Figure 3: **(a) Causal attention mask for joint attention.** It preserves causality while processing shared, left, and right inputs in parallel. **(b) TwinVLA joint attention mechanism.** The two VLMs share information, and the shared modality $(l, I_{\text{ego}})_t$ is further processed by MoE to more efficiently leverage both VLMs.

mechanism that dynamically routes shared tokens between the two VLM experts:

$$\text{MoE}(x) = w_{\text{left}} \cdot \text{FFN}_{\text{left}}(x) + (1 - w_{\text{left}}) \cdot \text{FFN}_{\text{right}}(x). \tag{3}$$

For calculating $w_{\text{left}}$, we add a linear layer that takes the embedding as input and outputs the weights via a softmax function. For other components like Projection, LayerNorm, we implement an output-averaging strategy inspired by task arithmetic (Tang et al., 2024). By processing inputs through both backbones and averaging their outputs, we functionally simulate a shared layer without physically merging parameters (see Figure 3b center). This efficient design reduces VRAM usage by 21%, enabling training with a batch size of 8 on a single 40 GB GPU.

**Attention re-weighting.** A potential side effect of introducing new arm-specific tokens is that the model's learned attention patterns can be disrupted, shifting focus away from the pretrained shared modalities. To mitigate this and preserve the valuable pretrained knowledge, we re-scale the attention scores for the shared modality (Algorithm 4). This maintains pretrained modality importance, allowing the model to bypass an initial adaptation phase and focus directly on the target task—a benefit evidenced by a lower initial loss and converged loss during fine-tuning.

## 5 EXPERIMENTS

In this paper, we propose TwinVLA to achieve strong bimanual manipulation performance with minimal bimanual data by fully leveraging a single-arm VLA pretrained on abundant single-arm data. Our empirical studies aim to answer the following questions:

- How does TwinVLA compare to state-of-the-art methods across diverse bimanual tasks, without any bimanual pretraining (Sections 5.2 and 5.3)?
- How quickly can TwinVLA adapt to new bimanual tasks (Section 5.4)?
- Does TwinVLA retain core VLA properties—language-following and robustness to unseen scenes and instructions (Sections 5.5 and 5.6)?
- How much does each key design choice contribute to overall performance (Section 5.7)?

### 5.1 COMPARED METHODS

We evaluate TwinVLA against three bimanual manipulation policies, each representing a different point in the design space.

- **RDT-1B** (Liu et al., 2024): This serves as our direct baseline. With a comparable size(1.2B vs. TwinVLA's 1.3B parameters), it represents the standard monolithic approach that requires

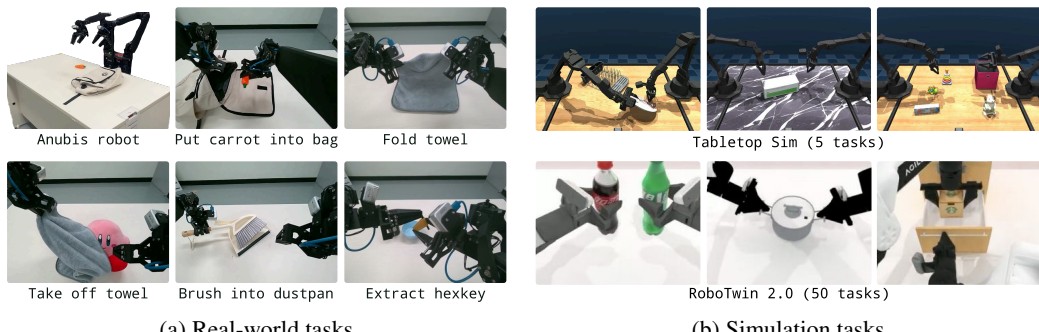

<table>
<tr><td>Anubis robot</td><td>Put carrot into bag</td><td>Fold towel</td><td>Tabletop Sim (5 tasks)</td></tr>
<tr><td>Take off towel</td><td>Brush into dustpan</td><td>Extract hexkey</td><td>RoboTwin 2.0 (50 tasks)</td></tr>
</table>

     (a) Real-world tasks               (b) Simulation tasks

Figure 4: **Experimental setups.** (a) We evaluate TwinVLA on five real-world bimanual tasks using an Anubis robot. (b) We further analyze TwinVLA on a large suite of simulation tasks: 5 tasks in Tabletop-Sim and 50 tasks in RoboTwin 2.0.

substantially larger resources ($\sim$2,400h data, $\sim$1,440 H100 days vs. $\sim$800h single-arm data, $\sim$25 H100 days).

- **$\pi_0$** (Black et al., 2024): We include this as an upper-bound, as this is 3.3B-parameter VLA trained on over 10K hours of proprietary robot data. Our goal is to assess how closely TwinVLA can approach this performance ceiling with far greater efficiency.

- **Diffusion Policy (DP)** (Chi et al., 2024a): This is a strong baseline method in low-data regime with 271M parameters, used to demonstrate the crucial benefits of pretraining.

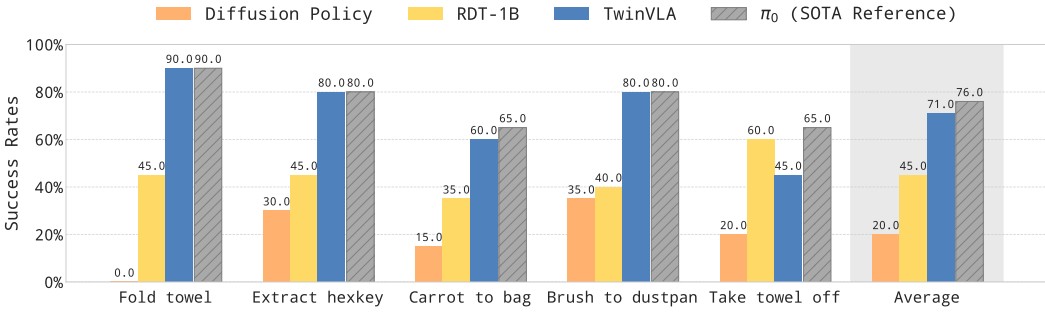

Figure 5: **Success rates on real-world tasks.** TwinVLA outperforms RDT-1B and DP on average. Moreover, TwinVLA shows comparable performance with $\pi_0$ while trained only on target data.

## 5.2 REAL-WORLD EXPERIMENTS

**Environment.** For real-world experiments, we use a dual-arm robot, Anubis (Kang et al., 2025), as shown in Figure 4a. Anubis has two 6 DoF arms with parallel-jaw grippers. The robot is equipped with two wrist-mounted cameras and a single ego-centric view camera.

**Tasks.** We design five long-horizon tabletop manipulation tasks, which require careful coordination and accurate motions: `Fold towel`, `Extract hexkey`, `Carrot to bag`, `Brush to dustpan`, and `Take towel off`, and one task set, `Put X into pot`. We collect 50 episodes for each task using absolute EEF control. Each method is fine-tuned for each task and evaluated with 20 rollouts.

**Results.** As presented in Figure 5, our model, TwinVLA, significantly outperforms RDT-1B. This achievement is remarkable considering the data disparity: TwinVLA is pretrained on just $\sim$800h of single-arm data, in contrast to RDT-1B's usage of a $\sim$2,400h dataset mixed with bimanual trajectories, which highlights the data efficiency of our approach. While DP's low performance confirms the necessity of pretraining, $\pi_0$ achieved the highest overall performance with significantly higher costs.

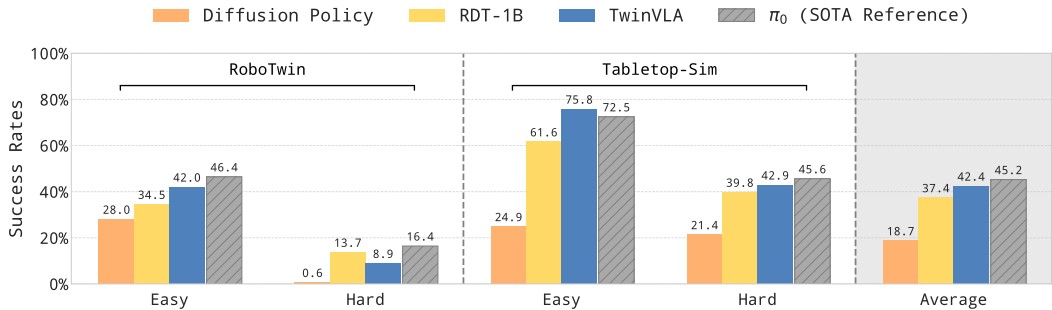

Figure 6: **Average success rates for diverse bimanual tasks.** Despite being pretrained solely on single-arm datasets, **TwinVLA** outperforms other methods except $\pi_0$.

## 5.3 SIMULATION EXPERIMENTS

**RoboTwin 2.0.** We use the RoboTwin 2.0 benchmark (Chen et al., 2025a), consisting of 50 bimanual tasks. Adhering to the official evaluation protocol, we fine-tune a model per task with 50 generated demonstrations and perform 100 test rollouts under both "Easy" and "Hard" settings. For Easy tasks, test scenes match the training data, but the instructions are novel. The Hard tasks introduce variations in texture, object position, and height. For compared methods, we use the results reported from RoboTwin 2.0 (Chen et al., 2025a).

**Tabletop-Sim.** To assess dexterous scenarios beyond tasks in RoboTwin, we develop Tabletop-Sim[1], a tabletop simulation environment based on `dm_control` (Tunyasuvunakool et al., 2020) and assets from ALOHA2 (Team et al., 2024) and GSO object dataset (Downs et al., 2022). We design 5 representative tasks that require precise bimanual coordination. Specifically, we define four single-tasks and one multi-task: `dish-drainer`, `handover-box`, `shoes-table`, `lift-box`, and `put X box into Y pot`. In the "Hard" tasks, we vary background textures and objects. We collect 50 episodes on each task using absolute EEF control, and fine-tune a model per task, and perform 500 evaluation rollouts for both "Easy" and "Hard" settings.

**Results.** The results in Figure 6 show the average success rates of TwinVLA and compared methods. DP, trained from scratch, shows the worst performance, highlighting the importance of pretraining. Once again, we observe that TwinVLA outperforms RDT-1B in most scenarios, except for the RoboTwin Hard tasks, and achieves comparable performance with $\pi_0$ by effectively leveraging single-arm data and modularity of bimanual manipulation. Notably, in Tabletop-Sim Easy tasks, TwinVLA even outperforms $\pi_0$, which is trained on an extensive corpus of high-quality bimanual pretraining data. This demonstrates TwinVLA's advantages in scenarios demanding higher dexterity and significant bimanual coordination.

## 5.4 DATA EFFICIENCY

TwinVLA exhibits data efficiency in two key aspects: pretraining and fine-tuning. For pretraining, it is efficient because it does not require supplemental bimanual data. For fine-tuning, it learns new tasks rapidly because its structural inductive bias facilitates the efficient transfer and application of its pretrained single-arm knowledge. We validate this efficiency in Tabletop-Sim Easy environment, comparing model's average success rates with varying amounts of demonstration data. As illustrated in Figure 7, TwinVLA exhibits a steep learning curve. Despite a modest start with 20 demonstrations, it quickly surpasses the performance of RDT with just 50 demonstrations, highlighting its exceptional data efficiency.

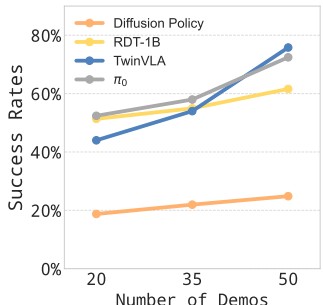

Figure 7: **Average success rates on the Tabletop-Sim Easy tasks.** Models are evaluated after fine-tuning with 20, 35, and 50 demonstrations.

---

[1]Our simulation setup is similar to the concurrent work Aloha-Sim, released by Google DeepMind (Google DeepMind, 2025).

## 5.5 POLICY ROBUSTNESS

One of the advantages of VLAs is their robustness to unseen situations and novel language instructions, thanks to pretraining. As shown in Figure 6, TwinVLA outperforms RDT-1B by $3.3\%$ even in the Hard setup of Tabletop-Sim, which involves different textures and objects.

The RoboTwin benchmark, both in the Easy and Hard setups, uses evaluation language instructions that are unseen during training. Here, TwinVLA again shows $7.48\%$ better performance than RDT-1B in the Easy setup. Although TwinVLA's performance on the RoboTwin Hard tasks is $3.72\%$ lower than that of RDT-1B, it still outperforms a non-pretrained Diffusion policy by $9.38\%$. This result demonstrates that TwinVLA possesses sufficient robustness as a bimanual VLA, even without being pretrained on large-scale bimanual manipulation data.

Table 1: Comparison of success rates for the `Fold towel` task in challenging scenes.

| Model | Low light | With distractors |
|---|---|---|
| RDT | 15.0% | 15.0% |
| $\pi_0$ | 40.0% | **60.0%** |
| **TwinVLA** | **45.0%** | 25.0% |

In Table 1, we additionally compared success rates in unseen real-world settings (see Figure 13)—specifically low-light and distractor-heavy environments—using the `Fold towel` task. TwinVLA is robust to lighting changes but less effective with distractors. Meanwhile, $\pi_0$ works robustly in both cases, and RDT-1B achieves the lowest success rates.

## 5.6 LANGUAGE FOLLOWING EVALUATIONS

A known challenge is that fine-tuning VLMs on robotic data can degrade their ability to faithfully follow nuanced instructions. We therefore evaluate how effectively our model preserves this core capability in a multi-task setting. We evaluated the "`Put X into pot`" task across both simulation and real-world settings. As observed in Figure 8a, TwinVLA outperforms both RDT-1B and $\pi_0$. We believe this performance stems from effectively preserving the knowledge acquired during single-arm pretraining through careful fine-tuning.

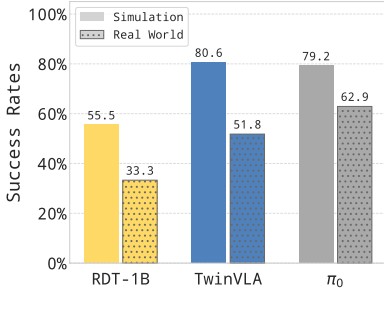
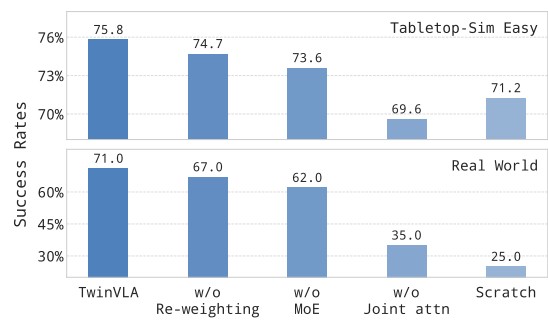

(a) Language task results

(b) Ablation results

Figure 8: **Language following task and ablation results.** (a) We evaluate average success rates on the language following tasks in the real world and Tabletop-Sim. (b) Ablation studies in the real world and Tabletop-Sim Easy tasks.

## 5.7 ABLATIONS

In this section, we conduct a sequential ablation study to analyze the cumulative impact of our key design choices on performance. Starting from the full TwinVLA model, we progressively remove each component in a specific order: first Attention Re-weighting, followed by MoE integration, and finally Joint Attention. This method reveals how performance degrades as each component of our architecture is stripped away. The results on our real-world and Tabletop-Sim Easy tasks are reported in Figure 8b.

**Attention re-weighting.** Removing the attention re-weighting mechanism (*w/o Re-weighting*) increased the initial fine-tuning loss by **40%** and decreased final performance by **1.1%** and **4.0%** in simulation and real world, respectively. This demonstrates that our re-weighting strategy successfully mitigates the input distribution shift between pretraining and fine-tuning.

**MoE integration.** Building on the previous ablation, we next remove the MoE integration (*w/o MoE*). This additional change increased the token sequence length by **28%** and increased VRAM usage by **21%**, making VLA training more burdensome. Surprisingly, it also further decreases the success rate by **1.1%** and **5.0%**, suggesting that MoE integration eliminates redundant processing of shared inputs while maintaining the performance.

**Joint attention.** Lastly, removing the joint attention mechanism (*w/o Joint attn*) causes the most significant additional performance drops of **4.0%** and **27.0%** in simulation and the real world, respectively. This impact is particularly pronounced in real-world tightly coupled bimanual tasks, confirming that joint attention is a critical mechanism for bimanual coordination.

**Effect of single-arm pretraining.** As a separate, foundational experiment, we assess the role of pretraining by training a model from scratch without OXE dataset (*Scratch*). This resulted in a **4.6%** performance drop in simulation and a stark **46.0%** in real world. This result confirms that effective cross-arm coordination is essential for bimanual manipulation and validates joint attention as the critical mechanism for achieving it in our model.

**Twin structure.** While we have confirmed that joint attention effectively connects the two modules, a crucial question remains: how does this approach compare to a monolithic model that is inherently unified from the start? To answer this, we revisit our comparison against RDT-1B, a monolithic model of a comparable 1.2B parameter size. The results are telling: TwinVLA outperforms RDT-1B by **26.0%** in the real world, **5.0%** in simulation, and **21.8%** in language-following tasks on average. This provides strong evidence that the inductive bias from the Twin Structure itself is highly beneficial for bimanual manipulation, validating our design choice over a monolithic approach.

## 6 LIMITATIONS

Generalization remains limited due to the visual disparity of two arms, which differs from the single-arm pretraining distribution. Future research into mechanisms that prevent this could address data scarcity by integrating diverse data, while also improving the model explainability and the better generalization ability to unseen tasks.

Moreover, we adopt absolute end-effector (EEF) pose control, as its embodiment-agnostic nature facilitates single-arm transfer, unlike DOF-specific joint positions. Future exploration of relative absolute actions (Chi et al., 2024b) or shared representations could further enhance transfer efficiency.

## 7 CONCLUSION

In this paper, we introduce TwinVLA, a data-efficient VLA model for bimanual manipulation. TwinVLA provides a new perspective on solving bimanual manipulation under scarce bimanual data by leveraging abundant single-arm datasets. From a small amount of bimanual demonstration data, TwinVLA learns to coordinate two copies of a SingleVLA pretrained on large-scale single-arm data via our proposed method. Through exhaustive experiments both in the real world and simulation, TwinVLA demonstrates its data-efficient learning of bimanual tasks compared to prior monolithic approaches. Beyond the bimanual setting, we believe this work serves as a blueprint for addressing inherent dataset imbalances across modalities. By illustrating how modular relationships can be exploited to bridge these data gaps, TwinVLA opens promising ways for other complex domains—such as mobile manipulation—thereby broadening the impact of large-scale robotic learning.

ACKNOWLEDGMENTS

This project was supported in part by Microsoft Research Asia and the Microsoft Accelerate Foundation Models Research (AFMR) grant program. This research was also supported by the Institute of Information & Communications Technology Planning & Evaluation (IITP) grants funded by the Korean Government (MSIT) (RS-2020-II201361, Artificial Intelligence Graduate School Program (Yonsei University); RS-2024-00436680, Global Research Support Program in the Digital Field Program), the Alchemist Project (RS-2024-00432143) funded by the Ministry of Trade, Industry & Energy (MOTIE, Korea), and the Electronics and Telecommunications Research Institute (ETRI)

grant funded by the Korean government (26ZR1100, Research on Intelligent Industrial Convergence). LG Electronics provided the Anubis robot, which was used for the experiments. The authors would like to thank Byeongjin Kang for the assistance with the preliminary experiments.

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

APPENDIX

## A    SINGLEVLA: EFFICIENT SINGLE-ARM POLICY DESIGN AND PRETRAINING

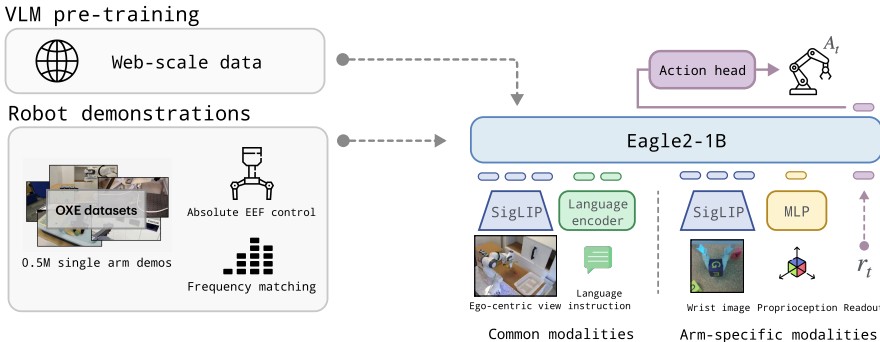

Figure 9: **Overview of SingleVLA architecture design and pretraining method.**

This section presents the design of the SingleVLA $\pi_{\text{single}}$. While SingleVLA follows established VLA conventions, our key novelty is a duplication strategy that enables the construction of TwinVLA. Prior 7B-scale models (Kim et al., 2024; 2025; Li et al., 2024a) are prohibitively large for such duplication, motivating a more efficient, lightweight Eagle2-1B (Li et al., 2025) based SingleVLA (Fig. 9). Since we do not use language head, the overall model size became 0.8B. To acquire generalizable knowledge, we pretrain SingleVLA on a ∼800h subset of the OXE mix, enabling transfer across diverse environments and embodiments. Pretraining ran for **120k** steps and took about **5 days** on a cluster with **5× H100** GPUs.

To ensure effective transfer to bimanual manipulation, it is crucial to choose an appropriate *action space*. Heterogeneous joint configurations across robots induce incompatible action spaces and complicate joint training. Prior work mitigates this with robot-specific decoders or high-dimensional zero-padded spaces (NVIDIA et al., 2025; Doshi et al., 2024; Octo Model Team et al., 2024; Black et al., 2024; Liu et al., 2024). Instead, we convert all actions into absolute end-effector (EEF) poses, providing a consistent, semantically meaningful representation across robots that naturally extends to bimanual control. For rotation, we adopt a 6D representation (Zhou et al., 2019), which is well suited for neural network learning.

### A.1    PRETRAINING

SingleVLA is pretrained on an OXE subset (∼800h); dataset composition and sampling rates appear in Table 2. We adopt the dataset loader from the OpenVLA (Kim et al., 2024) codebase and apply sampling according to the designated weights. Because some datasets (e.g., Kuka and BC-Z) include failed trajectories, we pre-process to retain only successful ones. Regarding the action space, we convert all actions to absolute EEF control with 6D rotations. We deliberately selected an absolute representation to mitigate the error accumulation and drift issues often amplified in high-frequency bimanual control. Unlike absolute joint positions, however, absolute EEF poses preserve the embodiment-agnostic property required for heterogeneous pretraining. We define these poses relative to the robot's base frame, resulting in a 10-Dimensional action space. We further apply *frequency matching* as described below.

**Frequency matching.**    Robotic datasets differ in control frequency, making fixed-length action-chunk prediction misaligned in real time. For example, a 20-step chunk spans ∼ 7 seconds in RT-1 (Brohan et al., 2022) (3 Hz) but only ∼ 1.3 seconds in DROID (Khazatsky et al., 2024) (15 Hz). Mixing low-frequency data like OXE (Open X-Embodiment Collaboration et al., 2024) with high-frequency datasets can degrade pretraining quality. Inspired by $\pi_0$-FAST (Pertsch et al., 2025), which uses DCT (Ahmed et al., 1974) to map 1-second actions into a consistent space, we perform frequency matching via interpolation: all datasets are resampled to 20 Hz, improving temporal alignment and transfer to high-frequency bimanual tasks.

Table 2: **SingleVLA pretraining datasets and sampling percentages.**

| Dataset | Sample Percentage |
| --- | --- |
| RT-1 (Brohan et al., 2022) | 24.49% |
| Kuka (filtered) (Yadav et al., 2024) | 12.40% |
| BridgeV2 (Walke et al., 2023) | 13.74% |
| Taco Play (Rosete-Beas et al., 2022) | 3.10% |
| Jaco Play (Dass et al., 2023) | 0.50% |
| Viola (Zhu et al., 2022a) | 1.00% |
| Berkeley Autolab UR5 (Chen et al., 2023) | 1.28% |
| Stanford Hydra (Belkhale et al., 2023) | 4.73% |
| Austin Buds (Zhu et al., 2022b) | 0.22% |
| NYU Franka Play (Cui et al., 2023) | 0.88% |
| FurnitureBench (Heo et al., 2023) | 2.40% |
| Austin Sailor (Nasiriany et al., 2022) | 2.33% |
| Austin Sirius (Liu et al., 2023c) | 1.84% |
| DLR EDAN (shared control) (Vogel et al., 2020; Quere et al., 2020) | 0.05% |
| UT Austin Mutex (Shah et al., 2023) | 2.38% |
| Berkeley FANUC manipulation (Zhu et al., 2023) | 0.82% |
| CMU Stretch (Bahl et al., 2023; Mendonca et al., 2023) | 0.16% |
| BC-Z (filtered) (Jang et al., 2021) | 7.90% |
| FMB (Luo et al., 2025) | 7.40% |
| Dobb-E (Shafiullah et al., 2023) | 1.50% |
| DROID (Khazatsky et al., 2024) | 10.70% |

## A.2 HYPERPARAMETERS AND COMPUTE

Table 3: **Key hyperparameters for TWINVLA training.**

| Hyperparameter | SingleVLA | TwinVLA |
| --- | --- | --- |
| Global batch size | 256 | 8 |
| Precision | BF16 | BF16 |
| Gradient clipping ($L_2$) | 1.0 | 1.0 |
| Learning rate | $1 \times 10^{-4}$ | $1 \times 10^{-4}$ |
| LR scheduler | cosine | cosine |
| Warm-up ratio | 0.01 | 0.05 |
| Total steps | 120k | 100k |
| Optimizer | AdamW | AdamW |
| Weight decay | $1 \times 10^{-5}$ | $1 \times 10^{-5}$ |
| Adam $\epsilon$ | $1 \times 10^{-8}$ | $1 \times 10^{-8}$ |
| Vision backbone frozen | true | true |
| Image augmentation | true | false |
| Action chunk size | 20 | 20 |
| Sampling step | 10 | 10 |

Table 3 summarizes training hyperparameters for SingleVLA and TwinVLA. SingleVLA pretraining used $5\times$ H100 GPUs for about 5 days. TwinVLA fine-tuning used $1\times$ L40S GPU for about 2 days.

## A.3 SINGLEVLA VLM ABLATION

We validate SingleVLA's VLM choice in the LIBERO (Liu et al., 2023a) environment using several VLMs. The LIBERO actions are converted to absolute EEF 6D control. Due to computational limits, we directly fine-tune the pretrained VLM checkpoints on LIBERO (i.e., without additional pretraining on LIBERO). Each model is evaluated with 500 rollouts per task suite under identical random seeds. Results are shown in Table 4.

Table 4: **Performance of different VLMs on LIBERO.**

| VLM | Spatial | Object | Goal | Long | Average |
|---|---|---|---|---|---|
| Qwen2VL-2B (Wang et al., 2024) | 80.4% | 88.6% | 83.8% | 43.0% | 73.9% |
| InternVL2.5-1B (Chen et al., 2025b) | 64.6% | 84.8% | 78.4% | 46.2% | 68.5% |
| Eagle2-1B (Li et al., 2025) | 73.4% | 85.4% | 90.8% | 46.6% | **74.0**% |

Although Qwen2VL is widely regarded as robust, Eagle2-1B achieves comparable or slightly better results while using roughly half the parameters and providing significantly faster inference. We therefore select **Eagle2-1B** as the VLM backbone for SingleVLA.

Table 5: **Performance of pretrained SingleVLA on LIBERO.**

| Method | Spatial | Object | Goal | Long | Average |
|---|---|---|---|---|---|
| SingleVLA (Eagle2-1B, no pretraining) | 73.4% | 85.4% | 90.8% | 46.6% | 74.0% |
| SingleVLA (pretrained) | **92.4**% | **94.5**% | **93.5**% | **63.7**% | **86.0**% |
| OpenVLA (Kim et al., 2024) | 84.7% | 88.4% | 79.2% | 53.7% | 76.5% |
| Octo (Octo Model Team et al., 2024) | 78.9% | 85.7% | 84.6% | 51.1% | 75.1% |

After pretraining SingleVLA with Eagle2-1B, we fine-tune it on LIBERO to assess single-arm capability. As shown in Table 5, the pretrained SingleVLA substantially improves performance and even surpasses the 7B model OpenVLA, indicating that the learned single-arm policy is both effective and sufficiently strong to benefit the bimanual policy.

## B    TRAINING DETAILS

Table 6: **Training hyperparameters for baseline models.**

| Method | # of params | Learning rate | Lr scheduler | Batch size | Training steps |
|---|---|---|---|---|---|
| TwinVLA | 1.3B | 1e-4 | cosine | 8 | 100k |
| RDT-1B | 1.2B | 1e-4 | constant | 8 | 100k |
| DP | 271M | 2e-5 | cosine | 8 | 100k |
| $\pi_0$ | 3.3B | 2.5e-5 | cosine | 8 | 100k |

We use the official implementation of RDT-1B. Diffusion Policy and $\pi_0$ are evaluated via the public LEROBOT release (Cadene et al., 2024), with two modifications for a fair comparison. First, the LEROBOT evaluation script normalized images differently from training; we corrected this to match the training pipeline.

All models are fine-tuned with the same number of steps and batch size so that the total number of training samples is consistent across methods. For learning rates, we began with each model's default and tuned within a similar compute budget. In practice, defaults worked well for DP and RDT-1B. For $\pi_0$, we observed better final returns by slowing the cosine decay; we therefore extended the LR schedule from 30k to 100k steps.

## C    TWINVLA DETAILS

### C.1    JOINT ATTENTION

The joint attention in TwinVLA is fundamentally almost identical to the implementation in the Mixture-of-Transformers (MoT) (Liang et al., 2024), but we applied attention-reweighting (Appendix C.3). While MoT has transformers for text, image, and speech inputs, in TwinVLA, the inputs for the left and right arms correspond to these.

---

**Algorithm 2** Joint Attention

---

1: **function** JOINTATTENTION($\{Q_m\}, \{K_m\}, \{V_m\}, M$)
2:     $Q, K, V \leftarrow$ Concatenate($\{Q_m\}, \{K_m\}, \{V_m\}$)     ▷ Concatenate modality-specific Q, K, V
3:     $S \leftarrow$ Softmax($(QK^\top/\sqrt{d_k}) + M$)     ▷ Apply causal joint mask M (Figure 3a)
4:     $S \leftarrow$ ApplyReweighting($S$)     ▷ Apply re-weighting (Algorithm 4)
5:     $A \leftarrow S \cdot V$     ▷ Calculate output A
6:     **return** $\{A_m\} \leftarrow$ Split($A$)     ▷ Split output $A$ into modality-specific $A_m$
7: **end function**

---

Furthermore, MoT requires an operation to group mixed inputs by modality and then restore their original order. However, this process is unnecessary in TwinVLA because the inputs are fed in a fixed sequence: left arm, then right arm. The detailed computation process is shown in Algorithm 2.

### C.2 MoE INTEGRATION

To enable sharing of the shared inputs between the two-arm models, we duplicated the entire VLM transformer. This necessitates different strategies for sharing the FFNs and the other components. This section details the strategy used for each component of the transformer.

**Feed-Forward Networks.** To share FFNs, we adopt the common approach of using a gating-based MoE. In standard MoE, multiple FFNs are included within a transformer, and a gating mechanism activates a subset for each input. In TwinVLA, the two VLMs act as distinct FFN experts.

Because shared inputs (e.g., ego-centric views or language prompts) may have asymmetric relevance for each arm, the gating mechanism learns how much each FFN should contribute to processing the shared input. This approach is widely used and has been shown to improve training stability and preserve information more effectively than simple averaging (Shazeer et al., 2017). We computed $w_{\text{left}}$ by applying a simple linear layer and softmax to the token embeddings.

**Other Components.** Beyond FFNs, elements such as layer normalization and projection layers also require integration. For these, we apply task arithmetic (Tang et al., 2024), merging the two VLMs via simple parameter averaging with weight $\lambda = 0.5$, elaborated Algorithm 3. This extends MoE-style computation to the full transformer architecture.

---

**Algorithm 3** Integration of other components

---

1: Let Projection$_b$ be projection layer from each backbone $b \in \{\text{left}, \text{right}\}$.
2: Let LayerNorm$_b$ be layernorm from each backbone $b \in \{\text{left}, \text{right}\}$.
3: **function** PROJ($X^m$)
4:     **if** $m =$ shared **then**
5:         $F^m \leftarrow 0.5 \cdot (\text{Projection}_{\text{left}}(X^m) + \text{Projection}_{\text{right}}(X^m))$     ▷ Task arithmetic
6:     **else**
7:         $F^m \leftarrow \text{Projection}_m(X^m)$
8:     **end if**
9:     **return** $F^m$
10: **end function**
11:
12: **function** NORM($X^m$)
13:     **if** $m =$ shared **then**
14:         $F^m \leftarrow 0.5 \cdot (\text{LayerNorm}_{\text{left}}(X^m) + \text{LayerNorm}_{\text{right}}(X^m))$     ▷ Task arithmetic
15:     **else**
16:         $F^m \leftarrow \text{LayerNorm}_m(X^m)$
17:     **end if**
18:     **return** $F^m$
19: **end function**

---

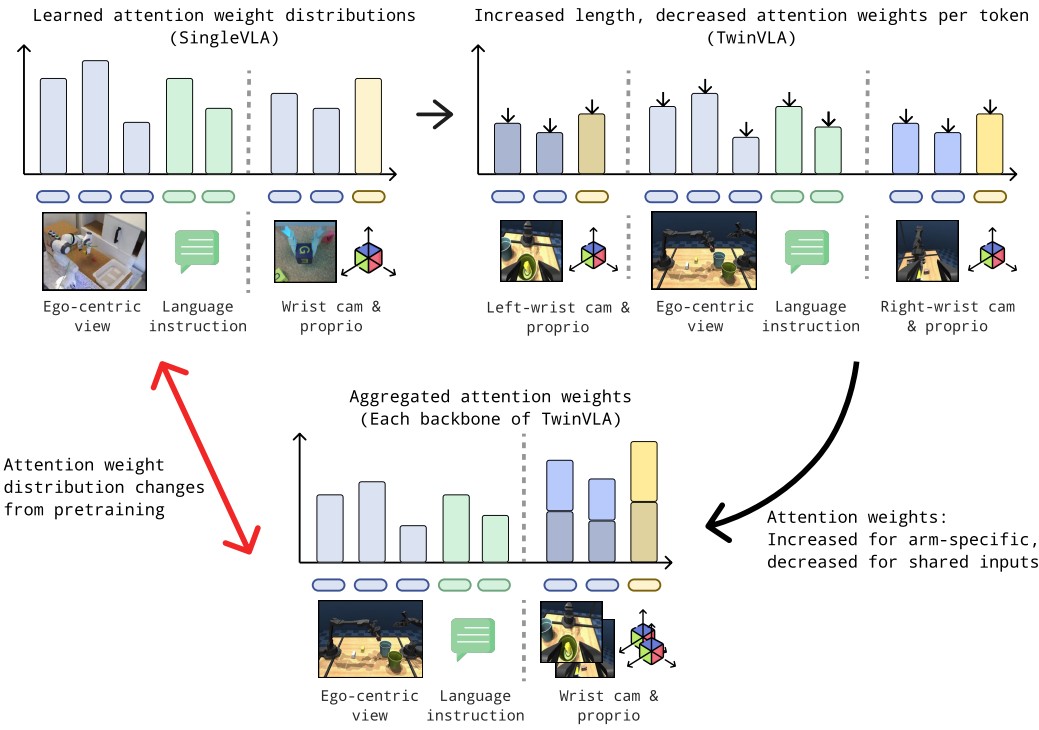

Figure 10: Due to the increased token length and softmax normalization, each VLM of TwinVLA refers to arm-specific inputs more than during pretraining, requiring the model to adapt.

---

**Algorithm 4** Attention Re-weighting

---

    **function** APPLYREWEIGHTING($\mathbf{A}$, $\alpha = 2$)
2:      Create mask $\mathbf{M_r} = (m \neq \text{shared})$       ▷ Create a mask for arm-specific inputs
      $\mathbf{A_{reweighted}} \leftarrow \mathbf{A} \odot (\mathbf{M_r} + \alpha \cdot \neg \mathbf{M_r})$ ▷ Apply scaling to attention weights using the mask
4:      $\mathbf{A_{reweighted}} \leftarrow \text{Normalize}(\mathbf{A_{reweighted}})$       ▷ Normalize the new weights
      **return** $\mathbf{A} + (\mathbf{A_{reweighted}} - \mathbf{A})$    ▷ Return weights as a residual update for gradient flow
6: **end function**

---

### C.3  ATTENTION RE-WEIGHTING

Attention re-weighting is a technique we employ to improve the efficiency of adapting a pretrained SingleVLA into a bimanual TwinVLA. Constructing TwinVLA involves adding a second set of arm-specific modality tokens. During operation, input tokens are processed by their corresponding arm's VLM backbone, pass through a joint attention layer, and then flow back to the individual VLMs. However, the softmax normalization within this joint attention layer presents a challenge. Although the total sequence length doubles, the number of tokens for shared inputs remains unchanged. Consequently, the proportion of attention allocated to these shared inputs is significantly diluted compared to the pretraining phase, creating a distribution shift for each VLM backbone's inputs, as illustrated in Figure 10.

This discrepancy requires greater adaptation effort for TwinVLA during fine-tuning on bimanual tasks. To address this, we introduce a simple re-weighting trick immediately after the attention scores are calculated. Specifically, we double the attention weights corresponding to the shared modality tokens and then re-normalize all weights to sum to one. This adjustment effectively restores the proportional attention each VLM backbone assigns to the shared inputs, aligning it with the pretraining conditions (see Figure 11). Applying this method reduced the initial fine-tuning loss by approximately 40%. While TwinVLA could learn bimanual manipulation without this technique, the required adaptation

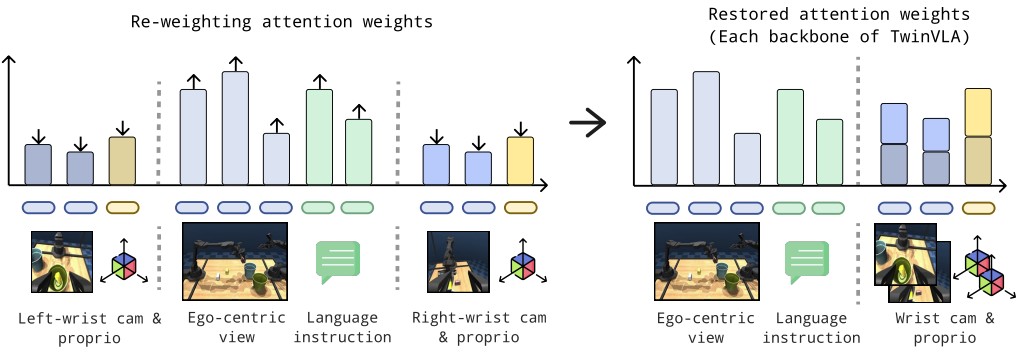

Figure 11: By re-weighting the attention weights, we can make each VLM refer to each modality identically to its pretraining stage, resulting in no adaptation and a lower initial loss.

period would be substantially longer. This simple trick makes the process significantly more efficient and faster. We illustrate our implementation with simple pseudocode in Algorithm 4.

# D    REAL-WORLD ROBOT EXPERIMENT DETAILS

## D.1    TASK DETAILS

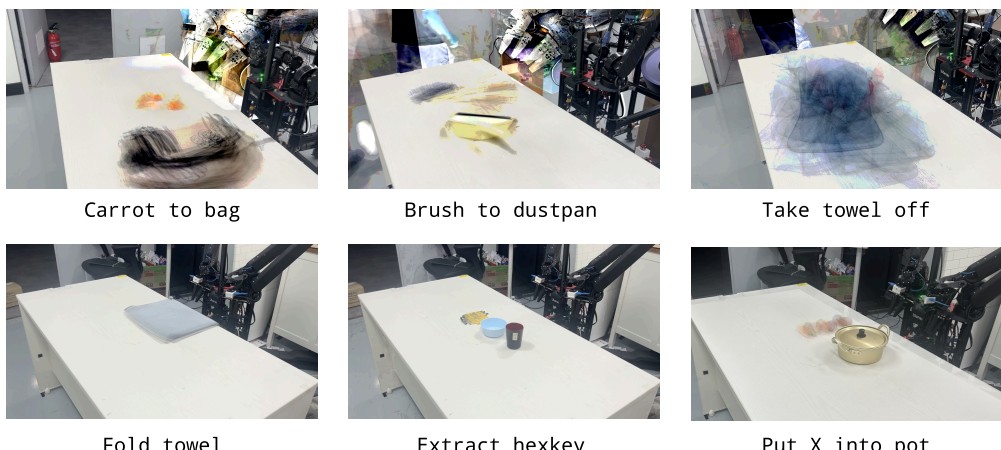

Figure 12: Initial distribution of each tasks in real-world.

To illustrate the diversity of initial configurations in our dataset, Figure 12 shows an overlay of the first frames from all 50 demonstrations. For each demonstration, the position and orientation of the objects were randomized, resulting in a unique starting setup.

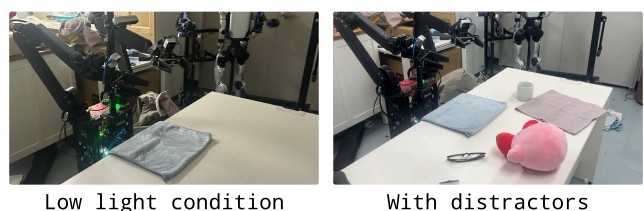

Figure 13: Challenging scene of Fold towel task.

Furthermore, to evaluate policy robustness in the real world, we tested the `Fold towel` task under more challenging conditions, such as reduced lighting and the presence of distractors. These scenarios are visualized in Figure 13.

## D.2 QUANTITATIVE RESULTS

Table 7: **Success rates for each model across all subtasks.** The best overall performance is highlighted in bold. As $\pi_0$ is included as an upper-bound, as this is excluded from this direct comparison.

| Task | Subtask | DP | TwinVLA | RDT-1B | $\pi_0$ |
|---|---|---|---|---|---|
| Fold towel | First fold | 0.00 | 1.00 | 0.90 | 1.00 |
| | Rotate | 0.00 | 1.00 | 0.85 | 1.00 |
| | Second fold | 0.00 | **0.90** | 0.45 | **0.90** |
| Extract hexkey | Pick up | 0.60 | 0.90 | 0.90 | 1.00 |
| | Extract | 0.35 | 0.80 | 0.55 | 0.90 |
| | Put into bowl | 0.30 | **0.80** | 0.45 | **0.80** |
| Carrot to bag | Pick up carrot | 0.50 | 1.00 | 0.75 | 0.85 |
| | Put carrot | 0.20 | 0.70 | 0.40 | 0.65 |
| | Close bag | 0.15 | 0.60 | 0.35 | **0.65** |
| Brush to dustpan | Move the brush | 0.70 | 1.00 | 1.00 | 1.00 |
| | Pick up the brush | 0.65 | 1.00 | 1.00 | 1.00 |
| | Put onto dustpan | 0.35 | **0.80** | 0.40 | **0.80** |
| Take towel off | Dragging | 0.40 | 0.90 | 0.80 | 0.95 |
| | Half off | 0.35 | 0.70 | 0.70 | 0.85 |
| | Entirely off | 0.20 | 0.45 | 0.60 | **0.65** |

We provide the quantitative results on real-world experiments in subtask-level detail in Table 7. The results reveal the main bottleneck in each long-horizon task. First, for the two tasks, `Fold towel` and `Extract hexkey`, requiring tightly coupled bimanual coordination, the phase where both arms meet to execute the action appears to be critical. The `Carrot to bag` task is challenging when inserting the carrot, which requires precisely opening the bag. The `Brush to dustpan` task's bottleneck is the high-precision insertion of the brush into the dustpan. Lastly, in `Take towel off`, the final unfolding is difficult—unlike the simple initial steps—as it requires a successful switch between the arms. In the next subsection, we show qualitative results from these specific bottleneck phases.

## D.3 QUALITATIVE RESULTS

Figure 14 presents qualitative results highlighting challenging situations for each task. A check mark was used when the model succeeded with a probability above 0.5, an X mark for probabilities below 0.3, and an exclamation mark icon for intermediate cases.

- `Carrot to bag`. $\pi_0$ showed the highest success rate, followed by TwinVLA, RDT, and DP. DP failed to interact meaningfully with the bag, especially struggling to grasp the cover properly. RDT failed to complete the task successfully, primarily due to its inability to accurately localize and grasp the bag's opening.
- `Brush to dustpan`. DP struggled either to grasp the brush itself or to successfully insert it. Interestingly, the RDT managed to grasp the brush well but lacked precision during the insertion. In this task, TwinVLA and $\pi_0$ demonstrated the same success rate.
- `Take towel off`. DP mostly failed to pull the doll from a distant position toward the center, while the other models succeeded in pulling it to the center but showed differences in towel removal. Both RDT and $\pi_0$ tended to successfully remove one side of the towel and then easily remove the other side as well. In contrast, TwinVLA struggled with removing the remaining part and repeated the same action. This is likely because the longer action chunk length of RDT and $\pi_0$ helped them overcome the multimodality challenge.

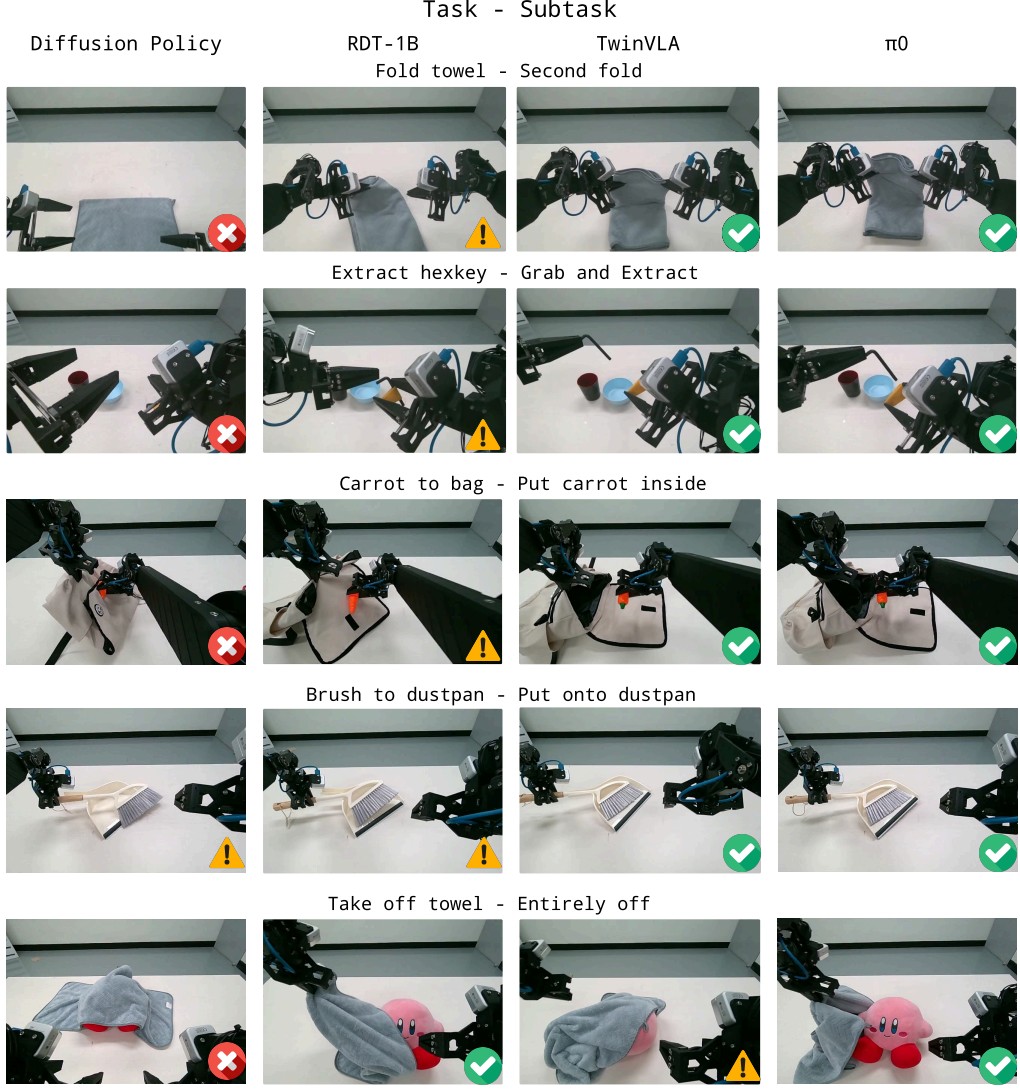

Figure 14: **Qualitative visualization of real world experiments.**

- `Fold towel.` $\pi_0$ and TwinVLA successfully completed the task. RDT also generally performed well, but occasionally failed to fully rotate the towel by 90 degrees, which caused downstream failures. DP experienced substantial difficulty with the fold-towel task and ultimately failed to solve it.

- `Extract hexkey.` $\pi_0$ and TwinVLA generally solved the task reliably. RDT performed the subtask of lifting the hexkey case well but often failed during extraction due to insufficient precision in grasping the hexkey once the case was lifted. DP failed both to reliably pick up the hexkey case and to extract the hexkey itself.

### D.4 ROBOT HARDWARE SPEC

We conduct our real-world experiments using a custom-built robot named Anubis. The platform features a teleoperation system inspired by the Mobile ALOHA setup (Fu et al., 2024). Each arm has 6 DoF and is equipped with a parallel gripper and a wrist-mounted camera. At the center of the robot, an Intel RealSense camera is mounted on a height-adjustable mechanism, serving as the ego-centric view camera. Details are described in Table 8. Anubis is equipped with a 3-wheel omni-directional base that supports planar locomotion; however, in this work, the mobility feature is not utilized.

Table 8: **Anubis Robot Hardware Specifications.**

| Component | Specification |
|---|---|
| Base Type | 3-wheel omni-directional chassis |
| Mobility DOF | 3 (X, Y, Yaw) |
| Arm DOF | 2 × (6 DOF + gripper) = 14 |
| Total Action Space | 17 DOF |
| Wrist Cameras | Intel RealSense D405 |
| Gripper | Parallel transparent gripper (hole design, ALOHA-style) |
| Power System | 3 × Greenworks 40V 5.0Ah batteries (PC, wheels & leader/follower) |
| Frame | 3D-printed custom components |

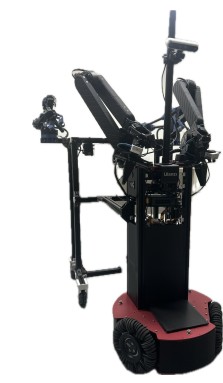

Figure 15: **The Anubis robot.**

# E    SIMULATION EXPERIMENT DETAILS

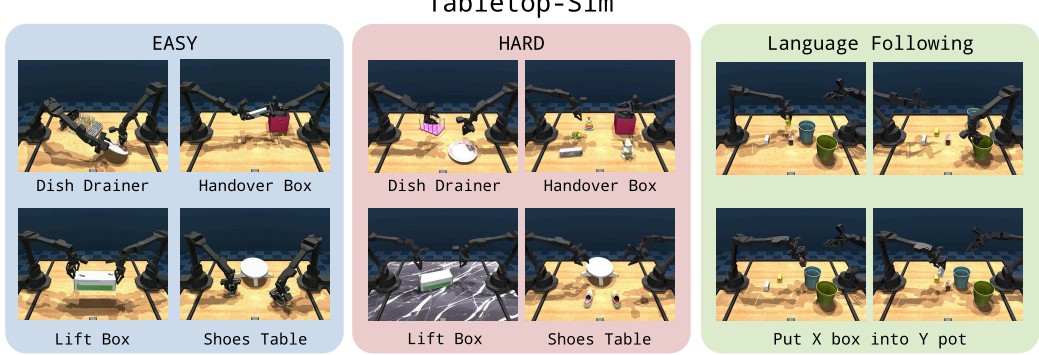

Figure 16: **Task list of Tabletop-Sim.**

## E.1    TABLETOP-SIM

To test bimanual policies in simulation, we developed Tabletop-Sim, a new benchmark specifically engineered to evaluate dexterous manipulation skills, in contrast to other benchmarks (Mu et al., 2025) that primarily focus on task diversity. The code are publicly available at `https://github.com/jellyho/Tabletop-Sim`. The benchmark comprises four single-task environments and one multi-task setup. Our task selection was guided by the taxonomy in DexMimicGen (Jiang et al., 2025), which categorizes bimanual tasks into: (1) parallel (two arms are doing separate tasks simultaneously), (2) coordinated (two arms are closely working together), and (3) sequential (one arm completes the task, and the other arm takes over) interactions. Using a custom controller similar to GELLO (Wu et al., 2024), we collected 50 demonstrations for each single-task and 60 for the multi-task environment.

The multi-task setup is a language-following task requiring the policy to place a specific box (out of three) into a designated pot (out of two) based on a language instruction. This task is designed to rigorously assess a model's instruction-following capabilities, as Vision-Language-Action (VLA) models often disregard instructions after fine-tuning.

Furthermore, to evaluate policy robustness, we established two difficulty settings for the four single-tasks. The original tasks are designated as the Easy setting, while a Hard variant for each task incorporates challenging variations such as different textures, object models, and the presence of distractor objects. Figure 16 presents snapshots of each task.

### E.2 QUANTITATIVE RESULTS

This section describes the detailed results for the simulation tasks. The results for **Tabletop-Sim** are listed in Table 9, while the results for the **RoboTwin 2.0** benchmark are in Table 10. For RoboTwin, the results for other baselines were referenced from the official benchmark results.

Although $\pi_0$ achieves the highest overall performance, this result is unsurprising considering its larger model size and pretraining dataset. Meanwhile, **TwinVLA** demonstrates consistently superior performance compared to **RDT-1B**, a model of a similar scale.

Table 9: **Performance comparison on the Tabletop-Sim benchmark.**

| | Tabletop-Sim | | | | | | | | |
| | Dish drainer | | Handover box | | Lift box | | Shoes table | | Put X cube in to Y pot |
| Model | Easy | Hard | Easy | Hard | Easy | Hard | Easy | Hard | |
|---|---|---|---|---|---|---|---|---|---|
| DP | 0.686 | 0.590 | 0.180 | 0.086 | 0.100 | 0.006 | 0.028 | 0.260 | - |
| RDT-1B | 0.810 | 0.780 | 0.694 | 0.508 | 0.300 | 0.076 | 0.660 | 0.192 | 0.555 |
| TwinVLA | **0.954** | **0.836** | 0.780 | **0.530** | 0.452 | 0.044 | **0.848** | 0.306 | **0.806** |
| PI-0 | 0.774 | 0.520 | **0.788** | 0.444 | **0.512** | **0.136** | 0.824 | **0.660** | 0.792 |

Table 10: **Success rates of TwinVLA for** 50 **bimanual tasks in RoboTwin 2.0.**

| Task Name | Easy | Hard | Task Name | Easy | Hard |
|---|---|---|---|---|---|
| adjust bottle | 0.97 | 0.35 | place can basket | 0.40 | 0.00 |
| beat block hammer | 0.77 | 0.10 | place cans plasticbox | 0.47 | 0.08 |
| blocks ranking rgb | 0.58 | 0.00 | place container plate | 0.77 | 0.04 |
| blocks ranking size | 0.03 | 0.00 | place dual shoes | 0.18 | 0.03 |
| click alarmclock | 0.33 | 0.01 | place empty cup | 0.50 | 0.01 |
| click bell | 0.58 | 0.13 | place fan | 0.34 | 0.00 |
| dump bin bigbin | 0.80 | 0.34 | place mouse pad | 0.50 | 0.00 |
| grab roller | 0.96 | 0.22 | place object basket | 0.48 | 0.03 |
| handover block | 0.17 | 0.00 | place object scale | 0.06 | 0.00 |
| handover mic | 0.84 | 0.02 | place object stand | 0.20 | 0.02 |
| hanging mug | 0.10 | 0.05 | place phone stand | 0.34 | 0.02 |
| lift pot | 0.87 | 0.07 | place shoe | 0.48 | 0.04 |
| move can pot | 0.45 | 0.05 | press stapler | 0.62 | 0.26 |
| move pillbottle pad | 0.32 | 0.02 | put bottles dustbin | 0.08 | 0.04 |
| move playingcard away | 0.61 | 0.35 | put object cabinet | 0.39 | 0.16 |
| move stapler pad | 0.11 | 0.00 | rotate qrcode | 0.54 | 0.03 |
| open laptop | 0.80 | 0.17 | scan object | 0.11 | 0.04 |
| open microwave | 0.03 | 0.01 | shake bottle horizontally | 0.96 | 0.55 |
| pick diverse bottles | 0.16 | 0.08 | shake bottle | 0.93 | 0.58 |
| pick dual bottles | 0.18 | 0.12 | stack blocks three | 0.00 | 0.00 |
| place a2b left | 0.27 | 0.05 | stack blocks two | 0.26 | 0.00 |
| place a2b right | 0.15 | 0.01 | stack bowls three | 0.77 | 0.15 |
| place bread basket | 0.11 | 0.03 | stack bowls two | 0.84 | 0.11 |
| place bread skillet | 0.20 | 0.01 | stamp seal | 0.16 | 0.01 |
| place burger fries | 0.67 | 0.13 | turn switch | 0.25 | 0.15 |

**Average**

| | | |
|---|---|---|
| Diffusion Policy | 0.280 | 0.006 |
| RDT-1B | 0.345 | 0.137 |
| TwinVLA | 0.420 | 0.089 |
| $\pi_0$ | 0.464 | 0.163 |

