# OpenReview forum: "TwinVLA: Data-Efficient Bimanual Manipulation with Twin Single-Arm Vision-Language-Action Models"
_ICLR.cc/2026/Conference — ICLR 2026 Poster_

### Official Review · Reviewer_5ANG · 2025-10-30

**Soundness:** 3
**Presentation:** 3
**Contribution:** 2
**Rating:** 6
**Confidence:** 5

**Summary:**

This paper proposes TwinVLA, a data-efficient framework that finetunes a pre-trained single arm VLA into a bimanual VLA without requiring any bimanual pretraining. The proposed method utilizes a joint attention mechanism of two identical pre-trained single-arm policies to adapt the bimanual action space. The experiment results show that TwinVLA achieves competible performance on downstream tasks compared to several bimanual VLAs that were pre-trained on large-scale bimanual manipulation data.

**Strengths:**

1. The single-arm policy can be trained to execute bimanual manipulation tasks without requiring any bimanual pre-training;
2. The proposed model can achieve competible performance compared to SOTA bimanual VLAs such as $\pi_0$ and RDT with much less training cost.

**Weaknesses:**

1. The paper has limited experimentation in real-world scenarios (only 3 tasks) and lacks comparisons with more bimanual VLA baselines;
2. There is a noticeable performance gap between TwinVLA and both $\pi_0$ and RDT in Hard tasks of the RoboTwin2.0 benchmark. More importantly, yet the study omits a detailed analysis of it.

**Questions:**

The paper indicates that TwinVLA performs poorly on the RoboTwin-Hard tasks. Is there a more in-depth analysis of this critical limitation? Furthermore, the authors also mention that TwinVLA exhibits weak generalization to unseen tasks. Do the aforementioned results reveal inherent weaknesses in this coordination mechanism's out-of-distribution generalization capability? Given that VLAs are expected to demonstrate strong generalization on unseen tasks, does this mean we still cannot avoid relying on large-scale bimanual pre-training data to achieve this goal?

---

> ### Author Response · Authors · 2025-11-22
>
> **W1.** Limited real-world scenarios.
>
> We have conducted three additional real-world evaluations to test the method under more challenging conditions. Specifically, we added two harder bimanual tasks that require tighter two-arm coordination (Fold towel and Extract hexkey) and one object-centric language-following task (Put X into pot).
>
> | Model | Fold towel | Extract hexkey | Put X into pot |
> | :--- | :---: | :---: | :---: |
> | DP  | 0.00%  | 30.0% | N/A |
> | RDT | 45.0% | 45.0% | 33.3% |
> | $\pi_0$ | **90.0%** | **80.0%** | **62.9%** |
> | TwinVLA (Ours) | **90.0%** | **80.0%** | 51.8% |
>
> TwinVLA achieved 90% and 80% success on Fold towel and Extract hexkey, respectively—matching $\pi_0$ (90%, 80%). On Put X into pot, TwinVLA reached 51.8%, outperforming RDT (33.3%) though remaining below $\pi_0$ (62.9%). These results are now reported and discussed in **Section 5.2 and 5.4**. Overall, they support that TwinVLA maintains strong performance on more coordination-heavy and language-conditioned real-world tasks, addressing the reviewer’s concern about limited real-world scenarios.
>
> &nbsp;
>
> **W2 & Q1.** Performance gap on Hard tasks & generalization
>
> We thank the reviewer for highlighting the performance gap in “Hard” tasks and for raising the important question regarding the limits of generalization. We agree that analyzing this limitation is crucial for understanding the boundaries of current VLA methods.
> To clarify our positioning, we distinguish between two primary goals in VLA research: *zero-shot generalization* (e.g., RT-2 [1]) and *efficient adaptation* (e.g., Octo [2]). TwinVLA is primarily  designed for the latter. Our core contribution is to demonstrate competitive adaptation efficiency on target bimanual tasks, without necessitating the prohibitive computational and data costs typically required for broad zero-shot capabilities. We now clarify this positioning more explicitly in the introduction.
>
> Regarding the specific limitations on "Hard" tasks, our analysis suggests that the performance drop is mainly due to a **visual distribution shift** in the shared input, rather than a failure of the coordination mechanism itself. While our modular architecture aims to preserve the pre-training input distribution for each backbone, the ego-centric view in a bimanual setup introduces a discrepancy: unlike single-arm pre-training data, this view simultaneously captures both arms. The visual presence of the "partner arm" thus acts as an out-of-distribution (OOD) factor.
>
> Our investigation of the Mixture-of-Experts (MoE) gating mechanism (described in Appendix D.3, Figure 15) reveals that the gates tend to route arm-specific information from ego-centric image tokens, but they still struggle to perfectly disentangle features when both arms are visible.This explains why performance degrades especially on visually challenging “Hard” tasks.
>
> Importantly, this does **not** imply that we must rely solely on large-scale bimanual pre-training to close this gap. Crucially, TwinVLA still exhibits superior generalization compared to models trained from scratch (e.g., Diffusion Policy), indicating that our method successfully leverages pre-trained priors. We believe the current limitations are addressable through relatively lightweight architectural solutions—such as arm-agnostic augmentation, stricter gating mechanisms, and additional training objectives—rather than brute-force scaling of bimanual data. Thus, TwinVLA offers a viable path to robust bimanual control that mitigates the dependency on scarce bimanual data.
>
> &nbsp;
>
> **References**
>
> [1] Brohan et al., RT-2: Vision-Language-Action Models Transfer Web Knowledge to Robotic Control, Arxiv, 2023
>
> [2] Octo Model Team, Octo: An Open-Source Generalist Robot Policy, RSS, 2024

---

> > ### Author Response · Authors · 2025-11-27
> >
> > Dear Reviewer 5ANG,
> >
> > We would like to kindly follow up on our earlier response to your review and hope that our clarifications have addressed your concerns.
> >
> > If you have any further questions or if there are remaining points you would like us to clarify, we would be very happy to discuss them with you before the discussion period ends.
> >
> > Thank you again for your time and thoughtful feedback.

---

### Official Review · Reviewer_Enff · 2025-10-31

**Soundness:** 2
**Presentation:** 3
**Contribution:** 3
**Rating:** 4
**Confidence:** 4

**Summary:**

This paper addresses the challenge of data scarcity in bimanual manipulation by proposing TwinVLA, a modular and data-efficient VLA framework. The core idea is to duplicate a VLA pre-trained on abundant single-arm data into two single-arm VLAs coupled via joint attention, with MoE routing, connecting them with a lightweight module for coordination. Experimental results demonstrate that this approach achieves strong performance by fine-tuning on only a small amount of bimanual data.

**Strengths:**

Modular twin design. Proposes TwinVLA, a clear, practical architecture that couples two single-arm pretrained VLAs via causal joint attention, enabling coordinated bimanual control while preserving per-arm specialization.

Efficiency mechanisms. Integrates MoE routing (and attention re-weighting on shared tokens) to avoid duplicate computation on shared inputs and to stabilize adaptation, yielding a favorable compute/memory profile.

Data efficiency. Achieves strong performance with only a small number of bimanual demonstrations for fine-tuning, offering a realistic path to leverage abundant single-arm corpora without large-scale bimanual pretraining.

**Weaknesses:**

Architectural advantage not causally established: The paper argues that the "twin structure" is superior to a "monolithic" one by comparing TwinVLA (1.3B) to RDT-1B (1.2B) . However, this comparison is confounded by major differences in their pre-training data and recipes. Without a size-matched monolithic VLA trained on exactly the same 0.5M single-arm pretraining data and the same 50 bimanual demos (with equal token/compute), the observed gains cannot be causally attributed to the twin.

Bimanuality insufficiently demonstrated in real-world tasks: The three real-world tasks appear predominantly sequential—two single-arm subtasks executed in series—rather than requiring concurrent, mutually constraining coordination. In the absence of tasks that necessitate simultaneous bilateral coupling (e.g., coordinated lifting/transport of a single large object), the evidence for learned genuine bimanual synergy remains weak.

**Questions:**

On the Language Following Evaluation (Sec 5.4): The paper claims TwinVLA's strong performance (80.6% vs. RDT-1B's 55.5%) is due to its architecture better preserving pre-trained language knowledge. How is a simpler confounding variable ruled out: that the baselines' (RDT-1B, Pi0) pre-training data mixtures were simply less effective at teaching these specific color-based instructions in the first place? Was any single-arm evaluation performed on the pre-trained checkpoints to confirm all models started with this capability before bimanual fine-tuning?

Aloha-Sim Easy may be too weakly bimanual. The w/o Joint attn variant (no cross-arm coordination) achieves 69.6%, only 6.2% below full TwinVLA (75.8%), suggesting these tasks may not require strong concurrent, mutually-constraining coordination. We request this 'w/o Joint attn' ablation be performed on the real-world and RoboTwin tasks to see if the coordination module proves more valuable there.

On 'Scratch' TwinVLA (71.2%) outperforming pre-trained RDT-1B (61.6%) on Aloha-Sim Easy: This is a counter-intuitive result, as one would expect the pre-trained RDT-1B to have an advantage. Could the authors analyze whether this finding points to: a) A superior inductive bias of the 'twin structure' itself for low-data bimanual learning, which is strong enough to overcome the lack of pre-training? b) A potential flaw, simplicity in the Aloha-Sim Easy benchmark?

---

> ### Author Response · Authors · 2025-11-22
>
> **W1.** Architectural advantage not causally established
>
> We thank Reviewer Enff for this critical assessment. We agree that isolating architectural variables is ideal for establishing causality. However, we respectfully argue that our comparison against RDT-1B actually provides **stronger causal evidence** for the superiority of the TwinVLA architecture than a controlled baseline would, because the experimental conditions were heavily weighted against our method.
>
> **1. Evidence from Resource Disparity**: The most significant confounding variable in this comparison is indeed the pre-training scale—but it favors the baseline. RDT-1B was trained on **2.8x more data** (1.4M mixed vs. 0.5M single-arm) and required approximately **50x more compute** (~1,440 vs. ~25 H100 days). The fact that TwinVLA outperforms RDT-1B in real-world, simulation, and language-following tasks **despite this massive resource deficit** logically isolates the "Twin" architecture as the source of efficiency. If our architecture were inferior or merely equal to the monolithic approach, it would not have overcome such a significant data and compute disadvantage.
>
> **2. Practical Constraints**: Regarding model size, creating a perfectly size-matched monolithic baseline (e.g., exactly 1.3B parameters) is non-trivial, as standard open-source VLM backbones are quantized in specific sizes (e.g., 1B, 2B, 7B). As noted in similar works (e.g., RDT-1B, $\pi_0$), perfect architectural control across diverse base models is often infeasible. We would still love to provide further evidence, so we are currently training a monolithic SingleVLA-2B model to compare with TwinVLA-1.3B. The results will come out in about three days.
>
> In summary, while the variables were not identical, the massive handicap under which TwinVLA succeeded provides compelling evidence that the performance gains are causally attributed to the architectural efficiency of the twin mechanism.
>
> &nbsp;
>
> **W2.** Extra evaluation on tasks with more coupled task.
>
> We conducted three additional real-world evaluations to test the method under more challenging conditions. We added two harder bimanual tasks that require closer two-arm coordination, (Extract hexkey, Fold towel) and one object-centric language-following task (Put X into pot).
>
>
> | Model | Fold towel | Extract hexkey |
> | :--- | :---: | :---: |
> | DP  | 0.00%  | 30.0% |
> | RDT | 45.0% | 45.0% |
> | $\pi_0$ | **90.0%** | **80.0%** |
> | TwinVLA (Ours) | **90.0%** | **80.0%** |
>
> TwinVLA achieved 90% and 80% on Fold towel and Extract hexkey, respectively—similar to $\pi_0$ (90%, 80%). These results are also presented in **Section 5.2 and 5.4**.
>
> &nbsp;
>
> **Q1.** Concern on color-based instruction setup.
>
> We thank Reviewer Enff for this insightful question regarding the language-following evaluation (Sec 5.4) and the potential for a confounding variable.
> To clarify, we have subsequently run additional real world language-following evaluations that require more semantic following rather than relying on color (Put X into pot).
>
> | Model | Put X into pot |
> | :--- | :---: |
> | RDT | 33.3% |
> | $\pi_0$ | **62.9%** |
> | TwinVLA (Ours) | 51.8% |
>
> TwinVLA continues to demonstrate stronger performance than RDT, while having some gap between $\pi_0$.We have added this result to Section 5.4 of the paper.
>
> We agree with the reviewer's premise that the pre-training data and recipes for TwinVLA (0.5M single-arm data), RDT-1B (1.4M mixed data), and $\pi_0$ (>10K hours proprietary data) are different. However, we respectfully suggest this difference highlights our method's *strength*, rather than a confounding factor. Both RDT-1B and $\pi_0$ are powerful, larger-scale models trained on significantly *more and more diverse data*. They are *expected* to have *greater* or at least more robust general language-following capabilities.
>
> The fact that TwinVLA—trained on only 0.5M single-arm data—so significantly outperforms RDT-1B (80.6% vs 55.5% in simulation and 51.8% vs 33.3% in real world) on this language-grounding task demonstrates that our modular architecture is exceptionally effective at *preserving* the VLM's pretrained language capabilities during fine-tuning. This, we argue, is achieved without the loss of generality that monolithic fine-tuning on mixed data might incur. This result strongly supports our central claim of efficient and effective knowledge transfer.

---

> ### Author Response · Authors · 2025-11-22
>
> **Q2.** Extra ablation on real world tasks.
>
> Thank you for your suggestion. We conducted additional ablation studies for the real-world tasks, and presented them in Section 5.7.
>
> As reported, joint attention and other components play an even more critical role in real-world tasks, creating substantial performance differences. Notably, the joint attention mechanism yielded a performance gap of **27.0%**. In the 'fold towel' task, where **bimanual coordination** is paramount, the agent struggled to solve the task without joint attention. This further validates that the joint attention mechanism—a core contribution of TwinVLA—remains highly effective.
>
> Additionally, while we considered conducting ablation studies on RoboTwin as requested, the extensive number of tasks and the associated resource constraints made it unfeasible. However, we believe that the results obtained from the real-world experiments provide sufficient evidence to support our claims.
>
> &nbsp;
>
> **Q3.** TwinVLA Scratch vs RDT-1B
>
> We thank the reviewer for highlighting this intriguing result. We agree that TwinVLA (Scratch) outperforming a pre-trained monolithic baseline is counter-intuitive. Regarding your analysis, we firmly believe this finding points to **(a) the superior inductive bias of the twin structure**, rather than a flaw in the benchmark.
>
> 1. Benchmark Validity:
>
> First, regarding the difficulty of Aloha-Sim Easy, we maintain that it remains a rigorous benchmark for evaluating fine-grained bimanual coordination. The fact that neither our scratch model nor the baselines achieve saturation (perfect scores) indicates that these tasks pose a significant control challenge. Unlike semantic-heavy tasks where pre-trained priors dominate, Aloha-Sim demands precise high-frequency inter-arm coordination, a regime where any pre-training does not always guarantee superior control dynamics.
>
> 2. Superior Inductive Bias:
>
> We interpret the success of TwinVLA (Scratch) (71.2%) over RDT-1B (61.6%) as evidence that our modular architecture provides a strong structural advantage that simplifies the optimization landscape. By explicitly disentangling the representation of each arm, TwinVLA can learn coordination mechanics efficiently even without pre-trained weights. This observation resonates with recent findings in the VLA literature, such as the $\pi_0$ paper (Figures 11 and 13), which notes that when an architecture is intrinsically well-suited to the control topology, the performance gap between scratch and pre-trained models can be surprisingly narrow in some cases.
>
> 3. The Necessity of Pre-training:
>
> However, this does not diminish the necessity of pre-training. We observe a clear performance gap between the Scratch model and TwinVLA even in simulation. **Crucially, this gap widens significantly when transitioning to the more challenging real-world experiments**. This trend confirms that while the "twin structure" provides an exceptional mechanism for coordination, the generalized representations from large-scale single-arm pre-training remain indispensable for handling complex, unstructured environments. Therefore, the validity of our approach lies in its ability to **efficiently transfer** these essential pre-trained capabilities to the bimanual setting, ensuring both coordination stability and robust generalization.

---

> > ### Author Response · Authors · 2025-11-27
> >
> > Dear Reviewer Enff,
> >
> > We would like to kindly follow up on our earlier response to your review and hope that our clarifications have addressed your concerns.
> >
> > If you have any further questions or if there are remaining points you would like us to clarify, we would be very happy to discuss them with you before the discussion period ends.
> >
> > Thank you again for your time and thoughtful feedback.

---

### Official Review · Reviewer_zQCq · 2025-11-01

**Soundness:** 3
**Presentation:** 2
**Contribution:** 3
**Rating:** 4
**Confidence:** 4

**Summary:**

TwinVLA proposes a modular approach for bimanual robotic manipulation by composing two pretrained single-arm vision-language-action models into a coordinated dual-arm system, avoiding the need for dedicated bimanual datasets. Instead of training a monolithic model on mixed data, it reuses single-arm policies and fuses them for coordinated control, improving data efficiency and generalization across both simulated and real-world settings. Experiments show that this compositional strategy can match or surpass prior large-scale bimanual methods without additional bimanual pretraining, suggesting a scalable and practical path to bimanual manipulation using existing public single-arm data.

**Strengths:**

- The evaluation is comprehensive and convincing, covering both simulation and real-world settings. The experiments span two simulation benchmarks and physical robot trials, with tasks designed to test both coordinated bimanual manipulation and asymmetric master-assistant roles.

- The method draws inspiration from neuroscience principles, providing an interesting conceptual link between biological motor coordination and modular robotic control.

- The approach significantly reduces VRAM and computational overhead compared to monolithic bimanual models, enabling training and deployment on more affordable hardware and lowering practical adoption barriers.

- The modular policy design offers strong scalability and flexibility, allowing pretrained single-arm expertise to be reused efficiently without requiring specialized or costly bimanual datasets.

**Weaknesses:**

- The core architecture and data-flow formulation of TwinVLA—arguably the central contribution—would benefit from a more formal and detailed presentation (*i.e.*, with equation and notations) in the main paper. Key notations and explanations are currently placed in the appendix, while the preliminaries and single-arm VLA review receive proportionally more emphasis. Streamlining those background sections could make room for a clearer, more rigorous exposition of the proposed twin-policy mechanism and coordination strategy.

- The real-world evaluation focuses largely on master–assistant settings, where one arm primarily aids the other. Including tasks that require tighter spatial and temporal coordination—such as hand-overs, bilateral grasping, or deformable-object manipulation (e.g., folding cloth by holding both corners)—would strengthen claims about general bimanual capability. These scenarios inherently require richer information exchange between the representations of the two arms, which may challenge the current decoupled design of TwinVLA and provide further insight into its scalability to fully interdependent control regimes.

- While the empirical results demonstrate benefits over monolithic bimanual models, the underlying mechanism for this advantage remains insufficiently explored. In my evaluation setting, this point is particularly relevant: as embodied datasets continue to scale, the performance gap between compositional and monolithic approaches may shift (Table 9, twinVLA vs pi0, btw, it's not a big deal). Consistent with the “bitter lesson,” it would be valuable to articulate whether the observed gains arise from modular inductive biases, improved optimization stability, or data-efficiency effects, rather than relying primarily on current empirical trends that may be influenced by the present limits of bimanual data availability. A deeper discussion here would strengthen confidence in the method’s long-term relevance and theoretical grounding.

I would be inclined to raise my score if the paper organization  are further improved.

**Questions:**

- L469 the absolute end-effector representation is inherently not frame-agnostic. Given that relative end-effector actions are typically more suitable for embodiment transfer, could the authors clarify why the relative formulation was not adopted here, and elaborate on the rationale behind choosing the absolute representation?

---

> ### Author Response · Authors · 2025-11-22
>
> **W1.** TwinVLA would benefit from a more formal and detailed presentation.
>
> We sincerely thank Reviewer zQCq for the constructive feedback. To directly address this concern, we have substantially revised the main paper to provide a clearer and more formal presentation of the TwinVLA architecture.
>
> To briefly summarize the key revisions made in response to this concern:
> * **Streamlined Background:** We condensed the general preliminaries on single-arm VLAs to reduce redundancy. Instead, we focused the background section on the theoretical groundwork essential for our method, specifically formally reviewing Mixture-Based Architectures (MoT and MoE).
> * **Formalized Method (Section 4):** Crucially, we have integrated the core computation logic—previously in the Appendix—directly into the main text. We now present **Algorithm 1** alongside **formal notations and equations** defining the TwinVLA. This explicitly details the exact data flow of the Joint Attention and MoE mechanisms.
> * **Clarified Positioning:** We refined the Related Work to explicitly compare TwinVLA with bimanual approaches like AnyBimanual [1] and InterACT [2], clarifying our unique position as an architecture-level transfer method.
>
> &nbsp;
>
> **W2.** The real world evaluation focuses largely on master-assistant settings.
>
> Thank you for the helpful suggestion. Following the reviewer’s comments, we conducted three additional real-world evaluations, and the results have been added to **Section 5.2 and 5.4**. This includes two more challenging bimanual tasks that require closer two-arm coordination (Fold towel, Extract hexkey).
>
> | Model | Fold towel | Extract hexkey ||
> | :--- | :---: | :---: | :---: |
> | DP  | 0.00%  | 30.0% |
> | RDT | 45.0% | 45.0% |
> | $\pi_0$ | **90.0%** | **80.0%** |
> | TwinVLA (Ours) | **90.0%** | **80.0%** |
>
> TwinVLA achieved 90% and 80% on Fold-Towel and Extract hexkey, respectively—matching $\pi_0$ (90%, 80%). These results show that **TwinVLA maintains strong performance on more coordination-heavy tasks**.
>
> &nbsp;
>
> **W3.** Discussion on TwinVLA’s long-term contribution.
>
> Thank you for this insightful question. Our perspective can be summarized along two main points:
> 1. while scalable methods are unquestionably valuable, there are **practical needs for making things work in a small-data regime**; and
> 2. in designing TwinVLA, we deliberately follow the spirit of the “bitter lesson” by avoiding heuristic, task-specific engineering and instead adopting a general architectural principle.
>
> &nbsp;
>
> **(1) Practical needs in small data regimes**
>
> Scalable monolithic policies benefit greatly from large datasets and computational resources However, real-world robotics applications often encounter limited data access. In these cases, simply scaling up a monolithic model and data is not always feasible.
>
> TwinVLA addresses this problem by leveraging pretrained single-arm VLAs, providing a favorable initialization that reduces optimization difficulty and enables stable learning with only modest amounts of bimanual demonstrations. This helps close the gap between what is theoretically possible at large scale and what is feasible under the practical constraints faced by many labs and deployments.
>
> **(2) Following the bitter lesson: minimal bias, not heuristic engineering**
>
> In designing TwinVLA, we aimed to respect—not contradict—the bitter lesson’s emphasis on avoiding narrowly hand-engineered solutions.
>
> Rather than introducing task-specific heuristics or specialized coordination rules that only work for individual tasks, we adopt a **task-agnostic architectural bias** that is broadly applicable across bimanual manipulation.
>
> Specifically, TwinVLA duplicates a pretrained single-arm VLA and links the two branches through a lightweight coordination interface (joint attention + MoE). This modular composition does not encode any task-specific or handcrafted structure; instead, it introduces a **domain-level inductive bias** for bimanual coordination. Because this structure is general, it can naturally scale with stronger backbones or larger bimanual datasets as they emerge.
>
> Thus, TwinVLA is not an alternative to scaling, but a design that **scales with scale**—providing a simple, general architectural principle that remains compatible with the computation- and data-driven trajectory highlighted by the bitter lesson.

---

> > ### Comment · Reviewer_zQCq · 2025-11-24
> > **Thank you for your valuable reply**
> >
> > Thank you for the thorough and thoughtful revisions which substantially address the key concerns I raised. With these improvements, I am willing to raise my score. I also encourage the authors to continue improving the clarity, conciseness, and overall writing quality as the paper further evolves.

---

> > > ### Author Response · Authors · 2025-11-24
> > >
> > > We sincerely thank Reviewer zQCq for re-evaluating our submission, raising the score, and acknowledging that our revisions substantially address the key concerns. We appreciate the suggestion to further improve clarity and conciseness, and will continue refining the paper accordingly.
> > >
> > > If there are any remaining issues that might influence your rating, we would be very grateful to hear them and address them.

---

> ### Author Response · Authors · 2025-11-22
>
> **Q1.** Rationale behind choosing absolute EEF action space.
>
> Recent literature, including Diffusion policy, ACT [3], $\pi_0$, OpenVLA-OFT [4], shows that absolute action spaces make a policy more stable. Similarly, our initial experiments with delta EEF actions revealed a critical stability issue (compounding errors) due to the combination of high-frequency control (>15Hz) and two independent sources of errors (left and right arms).
>
> For this reason, we opted for an **absolute representation** to ensure robust control while remaining more embodiment-agnostic than joint positions, which are strictly tied to kinematics. To maximize cross-embodiment compatibility within this absolute framework, we standardize all poses relative to the robot's base frame, a principled approach also adopted in recent works like X-VLA [5]. We have updated this rationale in **Appendix A.1**. We think that solving these action representation and drifting problems will be an interesting future work.
>
> &nbsp;
>
> **References**
>
> [1] Lu et al., AnyBimanual: Transferring Unimanual Policy for General Bimanual Manipulation, ICCV, 2025
>
> [2] Lee at al., InterACT: Inter-dependency Aware Action Chunking with Hierarchical Attention Transformers for Bimanual Manipulation, CoRL, 2024
>
> [3] Zhao et al., Learning Fine-Grained Bimanual Manipulation with Low-Cost Hardware, RSS, 2023
>
> [4] Kim et al., FIne-tuning Vision-Language-Action Models: Optimizing Speed and Success, Arxiv, 2025
>
> [5] Zheng et al., X-VLA: Soft-Promped Transformer as Scalable Cross-Embodiment Vision-Language-Action Model, Arxiv, 2025

---

### Official Review · Reviewer_yzn2 · 2025-11-04

**Soundness:** 3
**Presentation:** 2
**Contribution:** 3
**Rating:** 4
**Confidence:** 4

**Summary:**

This paper improves bimanual manipulation by combining single-arm pretraining with a Mixture-of-Experts (MoE) mechanism.
This approach enables the model to achieve performance close to state-of-the-art (SOTA) results while using significantly less pretraining data.

**Strengths:**

The motivation is sound, and the results demonstrate that it is a more efficient training approach.

The experimental results and supplementary materials provide additional details, and the experiments on weight changes in particular highlight the advantages of the proposed method.

**Weaknesses:**

1. The implementation details are not clearly described. The approach to handling bimanual manipulation appears somewhat similar to previous works, or at least it’s not evident how it fundamentally differs from other methods. More implementation details are needed from the authors to ensure an accurate evaluation.

[1] Anybimanual: Transferring Single-Arm Policy for General Bimanual Manipulation
[2] InterACT: Inter-dependency Aware Action Chunking with Hierarchical Attention Transformers for Bimanual Manipulation

2. It is unclear whether this fine-tuning paradigm, which is specifically designed for bimanual tasks, can be applied to robots with different degrees of freedom. There might be concerns regarding the generalizability of the method.

3. Will the authors conduct evaluations on more challenging tasks, such as tests under varying lighting conditions or complex backgrounds? I would like to understand the current limitations or upper bound of this fine-tuning approach.

**Questions:**

Please refer to the weaknesses.

---

> ### Author Response · Authors · 2025-11-22
>
> **Q1.** The implementation details are not clearly described.
>
> We thank Reviewer yzn2 for the feedback. In the original submission, the full implementation details were provided in the Appendix, but we agree that they were not sufficiently emphasized in the main text, and that the differences from the prior work were not clearly highlighted.
>
> To address this, we have substantially clarified and surfaced the implementation details:
>
> First, we have moved the core computation logic from the Appendix into the main paper. We now present **Algorithm 1**, together with **formal notations and equations** that defines TwinVLA. This explicitly specifies the exact data flow of the Joint Attention and MoE mechanisms, making the architecture and training procedure directly accessible without having to rely on the appendix alone.
>
> Second, we have revised the Related Work and Method sections to more explicitly contrast TwinVLA with AnyBimanual [1] and InterACT [3]. In particular, we clarify that TwinVLA focuses on efficiently transferring **existing** single-arm VLAs to a bimanual VLA without introducing additional high-level modules or restricting the backbone class. This is in contrast to AnyBimanual [1], which requires an additional ‘high-level skill manager’, complex 3D input management, and PerAct-based backbones [2]. InterACT [3] also requires a ‘hierarchical attention transformer’, which is not directly applicable to existing VLM backbones (i.e. lacks large-scale validation) and has shown marginal improvements.
>
> Furthermore, we added a brief explanation of mixture-based methods (MoT and MoE) to make the architectural choices behind TwinVLA easier to follow in Section 3.3. With these revisions, the architecture, data flow, and distinctions from prior works are now clearly described in the main paper.
>
> &nbsp;
>
> **Q2.** Can this fine-tuning paradigm be applied to robots with different DOFs?
>
> Yes. Fine-tuning within our modular architecture is agnostic to the specific degrees of freedom of each robot component. Our modular design therefore offers a scalable solution for multi-domain settings.
>
> For example, mobile manipulation can be approached by (1) combining a pre-trained navigation model (e.g., GNM [4]) with a pre-trained manipulation model (e.g., SingleVLA or TwinVLA), and (2) fine-tuning the entire model using only a few mobile manipulation data, rather than requiring abundant mobile manipulation demonstrations. This effectively addresses the inherent data asymmetry across modalities.
>
> By enabling independent pre-training for each component, our framework provides a practical blueprint for extending the same fine-tuning paradigm beyond bimanual arms to general robotic systems with diverse DoFs.

---

> ### Author Response · Authors · 2025-11-22
>
> **Q3.** Will the authors conduct extra evaluation on more challenging tasks?
>
> Thank you for the helpful suggestion. Following the reviewer’s comments, we conducted three additional real-world evaluations, and the results have been added to **Section 5.2 and 5.4**. This includes two more challenging bimanual tasks that require closer two-arm coordination (Fold towel, Extract hexkey) and one object-centric language-following task (Put X into pot).
>
>
>
> | Model | Fold towel | Extract hexkey | Put X into pot |
> | :--- | :---: | :---: | :---: |
> | DP  | 0.00%  | 30.0% | N/A |
> | RDT | 45.0% | 45.0% | 33.3% |
> | $\pi_0$ | **90.0%** | **80.0%** | **62.9%** |
> | TwinVLA (Ours) | **90.0%** | **80.0%** | 51.8% |
>
> TwinVLA achieved 90% and 80% on Fold towel and Extract hexkey respectively—matching $\pi_0$ (90%, 80%). On Put X into pot, TwinVLA reached 51.8%, outperforming RDT (33.3%) while remaining below $\pi_0$ (62.9%). These results show that TwinVLA maintains strong performance on more coordination-heavy tasks.
>
> Furthermore, as suggested, we performed robustness evaluations under (i) low-light conditions and (ii) visual distractors on the Fold towel task, and included these results in **Section 5.6**. In summary, TwinVLA shows more robust performance on different lighting conditions than $\pi_0$, while suffering from distractors.
>
>
> | Model | Low light | With distractors |
> | :--- | :---: | :---: |
> | RDT | 15.0% | 15.0% |
> | $\pi_0$ | 40.0% | **60.0%** |
> | TwinVLA (Ours) | **45.0%** | 25.0% |
>
> In the low-light setting, **TwinVLA achieved a success rate of 45% over 20 trials**, outperforming RDT (15%) and $\pi_0$ (40%). However, performance degraded to **25.0%** in scenarios with visual distractors, whereas $\pi_0$ achieved **60.0%**. This limitation aligns with our earlier findings in the **RoboTwin Hard** benchmark, indicating a constraint in TwinVLA’s generalization capabilities. We attribute this to the **distribution shift in the ego-centric view**; specifically, the presence of two arms—which were unseen during the single-arm backbone pre-training—introduces a visual disparity that our current information transfer mechanism does not fully mitigate. Addressing this specific visual domain gap remains a key direction for future work.
>
> &nbsp;
>
> **References**
>
> [1] Lu et al., AnyBimanual: Transferring Unimanual Policy for General Bimanual Manipulation, ICCV, 2025
>
> [2] Shridhar et al., Perceiver-Actor: A Multi-Task Transformer for Robotic Manipulation, CoRL, 2022
>
> [3] Lee at al., InterACT: Inter-dependency Aware Action Chunking with Hierarchical Attention Transformers for Bimanual Manipulation, CoRL, 2024
>
> [4] Shah et al., GNM: A General Navigation Model to Drive Any Robot, ICRA, 2023

---

> > ### Author Response · Authors · 2025-11-27
> >
> > Dear Reviewer yzn2,
> >
> > We would like to kindly follow up on our earlier response to your review and hope that our clarifications have addressed your concerns.
> >
> > If you have any further questions or if there are remaining points you would like us to clarify, we would be very happy to discuss them with you before the discussion period ends.
> >
> > Thank you again for your time and thoughtful feedback.

---

### Author Response · Authors · 2025-11-22

We sincerely thank the reviewers for their thoughtful and constructive feedback. We have addressed all comments and revised the paper accordingly, with substantial updates highlighted in yellow.

**(1)** New real-world experiments

- As requested by the reviewers, we conducted new real-world evaluations that include more challenging bimanual tasks and more semantically grounded language-following tasks. These results have been incorporated into **Section 5.2** and **Section 5.4**.

- We performed a comprehensive ablation study of TwinVLA on an expanded set of real-world tasks. The corresponding analyses are now included in **Section 5.7**.

- We further evaluated policy robustness under more challenging real-world conditions (e.g., low light, distractors). These results have been added to **Section 5.6**.

- To avoid confusion with Google’s **Aloha-Sim [1]** environment, we renamed our simulation environment from **Aloha-Sim** to **Tabletop-Sim**.

**(2)** Paper revision

- To address the reviewers’ concerns regarding insufficient implementation details, we moved detailed descriptions of the TwinVLA framework from the appendix to the main paper: **Joint Attention** and **Mixture of Experts (MoE)**—throughout the **Preliminaries** and **TwinVLA** sections, as well as in **Appendix C (TwinVLA Details)**.

- We compared our method with additional prior works (AnyBimanual [2], InterACT [3], and Bi-vla [4]) and clearly positioned our method in the **Related Work** section.


**Reference**

[1] Google DeepMind. Aloha-Sim: A collection of tabletop tasks in MuJoCo. GitHub, 2025. Available at: https://github.com/google-deepmind/aloha_sim

[2] Lu et al., AnyBimanual: Transferring Unimanual Policy for General Bimanual Manipulation, ICCV, 2025

[3] Lee at al., InterACT: Inter-dependency Aware Action Chunking with Hierarchical Attention Transformers for Bimanual Manipulation, CoRL, 2024

[4] Kobayashi et al., Bi-vla: Bilateral control-based imitation learning via vision-language fusion for action generation, Arxiv, 2025

---

### Meta-Review · Area_Chair_Bk3b · 2026-01-10

**Summary:**

TwinVLA’s contribution is to enable data-efficient bimanual manipulation by composing pretrained single-arm VLAs, eliminating the need for large-scale bimanual pretraining. This approach enables the model to achieve performance close to state-of-the-art (SOTA) results while using significantly less pretraining data.

**Reviewer Concerns:**

The major review concerns are summarized as follows:

- More challenging new real-world experiments. This is raised by several reviewers.0
- While the empirical results demonstrate benefits over monolithic bimanual models, the underlying mechanism for this advantage remains insufficiently explored.
- Architectural advantage not causally established.

The rest of the concerns raised by reviewers (clarity, more explaination on the technical implementation), are addressed by authors in great details. AC just omit these concerns to write them here.

**Reviewer Scores:**

Authors did a pretty good job addressing most concerns. Although the preliminary score leans negative, most review concerns are gentle and mild. Notably, authors have added new real-world experiments and polish the paper in great extent.

AC read the paper, revision, rebuttal and review comments; and believe the paper has been polished in good shape. Please revise the manuscript and incorporate all review comments before camera-ready version.

---

### Decision · Program_Chairs · 2026-01-26

Accept (Poster)